# A Machine Learning Examination of Hydroxyl Radical Differences Among Model Simulations for CCMI-1

Julie M. Nicely[1,2], Bryan N. Duncan[2], Thomas F. Hanisco[2], Glenn M. Wolfe[2,3] Ross J. Salawitch[1,4,5], Makoto Deushi[6], Amund S. Haslerud[7], Patrick Jöckel[8], Béatrice Josse[9], Douglas E. Kinnison[10], Andrew Klekociuk[11,12], Michael E. Manyin[2,13], Virginie Marécal[9], Olaf Morgenstern[14], Lee T. Murray[15], Gunnar Myhre[7], Luke D. Oman[2], Giovanni Pitari[16], Andrea Pozzer[17], Ilaria Quaglia[16], Laura E. Revell[18], Eugene Rozanov[19,20], Andrea Stenke[19], Kane Stone[21,22], Susan Strahan[2,23], Simone Tilmes[10], Holger Tost[24], Daniel M. Westervelt[25,26], and Guang Zeng[14]

[1]Earth System Science Interdisciplinary Center, University of Maryland, College Park, MD, USA.
[2]NASA Goddard Space Flight Center, Greenbelt, MD, USA.
[3]Joint Center for Earth Systems Technology, University of Maryland Baltimore County, Baltimore, MD, USA.
[4]Department of Atmospheric and Oceanic Science, University of Maryland, College Park, MD, USA.
[5]Department of Chemistry and Biochemistry, University of Maryland, College Park, MD, USA.
[6]Meteorological Research Institute (MRI), Tsukuba, Japan.
[7]Center for International Climate and Environmental Research-Oslo (CICERO), Oslo, Norway.
[8]Institut für Physik der Atmosphäre, Deutsches Zentrum für Luft- und Raumfahrt (DLR), Oberpfaffenhofen, Germany.
[9]CNRM UMR 3589, Météo-France/CNRS, Toulouse, France.
[10]National Center for Atmospheric Research, Boulder, CO, USA.
[11]Antarctica and the Global System Program, Australian Antarctic Division, Kingston, Australia.
[12]Antarctic Climate and Ecosystems Cooperative Research Centre, Hobart, Australia.
[13]Science Systems and Applications, Inc., Lanham, MD, USA.
[14]National Institute of Water and Atmospheric Research (NIWA), Wellington, New Zealand.
[15]Department of Earth and Environmental Sciences, University of Rochester, Rochester, NY, USA.
[16]Department of Physical and Chemical Sciences, Universitá dell'Aquila, L'Aquila, Italy.
[17]Max-Planck-Institute for Chemistry, Air Chemistry Department, Mainz, Germany.
[18]School of Physical and Chemical Sciences, University of Canterbury, Christchurch, New Zealand.
[19]Institute for Atmospheric and Climate Science, ETH Zürich (ETHZ), Zürich, Switzerland.
[20]Physikalisch-Meteorologisches Observatorium Davos – World Radiation Center (PMOD/WRC), Davos, Switzerland.
[21]School of Earth Sciences, University of Melbourne, Melbourne, Australia.
[22]Massachusetts Institute of Technology, Cambridge, MA, USA.
[23]Universities Space Research Association, Columbia, MD, USA.
[24]Institute for Atmospheric Physics, Johannes Gutenberg University, Mainz, Germany.
[25]Lamont-Doherty Earth Observatory, Columbia University, Palisades, New York, USA.
[26]NASA Goddard Institute for Space Studies, New York, NY, USA.

*Correspondence to*: Julie M. Nicely (julie.m.nicely@nasa.gov)

**Abstract.** The hydroxyl radical (OH) plays critical roles within the troposphere, such as determining the lifetime of methane (CH$_4$), yet is challenging to model due to its fast cycling and dependence on a multitude of sources and sinks. As a result, the reasons for variations in OH and the resulting methane lifetime ($\tau_{CH4}$), both between models and in time, are difficult to diagnose. We apply a neural network (NN) approach to address this issue within a group of models that participated in the Chemistry-Climate Model Initiative (CCMI). Analysis of the historical specified dynamics simulations performed for CCMI

indicates that the primary drivers of $\tau_{CH4}$ differences among ten models are the flux of UV light to the troposphere (indicated by the photolysis frequency $JO^1D$), the mixing ratio of tropospheric ozone ($O_3$), the abundance of nitrogen oxides ($NO_x \equiv NO+NO_2$), and details of the various chemical mechanisms that drive OH. Water vapor, carbon monoxide (CO), the ratio of $NO:NO_x$, and formaldehyde (HCHO) explain moderate differences in $\tau_{CH4}$, while isoprene, methane, the photolysis frequency of $NO_2$ by visible light ($JNO_2$), overhead ozone column, and temperature account for little-to-no model variation in $\tau_{CH4}$. We also apply the NNs to analysis of temporal trends in OH from 1980 to 2015. All models that participated in the specified dynamics historical simulation for CCMI demonstrate a decline in $\tau_{CH4}$ during the analysed timeframe. The significant contributors to this trend, in order of importance, are tropospheric $O_3$, $JO^1D$, $NO_x$, and $H_2O$, with CO also causing substantial interannual variability in OH burden. Finally, the identified trends in $\tau_{CH4}$ are compared to calculated trends in the tropospheric mean OH concentration from previous work, based on analysis of observations. The comparison reveals a robust result for the effect of rising water vapor on OH and $\tau_{CH4}$, imparting an increasing and decreasing trend of about 0.5 % decade$^{-1}$, respectively. The responses due to $NO_x$, ozone column, and temperature are also in reasonably good agreement between the two studies.

## 1    Introduction

The hydroxyl radical (OH) is a key species of interest for numerous tropospheric chemistry studies over the past several decades (e.g., Prinn et al., 1987, 1992; Spivakovsky et al., 2000; Montzka et al., 2011; Prather et al., 2012; Holmes et al., 2013; Murray et al., 2013; Naik et al., 2013; Voulgarakis et al., 2013; McNorton et al., 2016; Rigby et al., 2017; Turner et al., 2017). As a result of its role as the primary daytime oxidant in the lower atmosphere, OH determines how quickly many tropospheric gases and aerosols degrade or transform chemically. Notably, loss of atmospheric methane ($CH_4$) is dominated by its reaction with OH. Uncertainties in the abundance of OH at the global scale, coupled with source terms of methane that are difficult to quantify, have driven disagreement in the causes of recent variations in the methane growth rate (Nisbet et al., 2019; Turner et al., 2019). As a key element in the methane budget, tropospheric OH must be studied further to clarify its present-day abundance as well as its variability over time.

Numerous studies have sought to constrain the OH abundance and resulting methane lifetime ($\tau_{CH4}$) using observations, global atmospheric models, and combinations of the two. Historically, chemical inversion of methyl chloroform (MCF: $CH_3CCl_3$) comprised the primary method capable of gleaning information about global-scale OH burdens (Lovelock, 1977; Prinn et al., 1987; Ravishankara and Albritton, 1995; Krol et al., 1998; Montzka et al., 2000; Spivakovsky et al., 2000; Bousquet et al., 2005), though additional species that are lost by reaction with OH were also tested for this purpose (Weinstock, 1969; Singh, 1977; Miller et al., 1998; Jöckel et al., 2002; Nisbet et al., 2016, 2019; Liang et al., 2017). Models have likewise been relied upon to derive tropospheric OH abundance and its evolution. Stevenson et al. (2006) found a large spread in $\tau_{CH4}$ (6.3 to 12.5 years) from a suite of atmospheric chemistry models in an analysis performed more than a decade ago. Seven years later, the

Atmospheric Chemistry and Climate Model Intercomparison Project (ACCMIP) generated both historical (Naik et al., 2013) and future (Voulgarakis et al., 2013) simulations from numerous chemistry-climate models, revealing still large discrepancies not only in present-day $\tau_{CH4}$ (with values ranging from 7.1 to 14.0 years) but also in how $\tau_{CH4}$ is expected to vary through the year 2100 given common emissions scenarios. Note that, here and throughout, $\tau_{CH4}$ refers to the lifetime of methane due to reaction with tropospheric OH only. Most recently, the confluence of observations with advanced modelling techniques have enabled sophisticated analyses of global OH (Prather et al., 2012; Holmes et al., 2013; McNorton et al., 2016; Rigby et al., 2017; Turner et al., 2017). Despite the advent of numerous observing systems for species with some bearing on OH chemistry in the last several decades, it is widely acknowledged that current observations are insufficient to unambiguously derive current trends in OH (Prather & Holmes, 2017; Turner et al., 2017, 2019; Nisbet et al., 2019).

While global models are insufficient for clarifying the outstanding questions regarding OH and $\tau_{CH4}$ on their own, they can serve as valuable testbeds in which to evaluate the factors influencing OH chemistry. At the global scale, the dominant reactions responsible for producing, cycling, and sequestering OH (see, e.g., Spivakovsky et al. (2000)) are well characterized and represented, to varying degrees of explicitness, in modern chemical mechanisms. Despite general consensus on the immediate drivers of OH chemistry, large differences in OH can manifest due to infrequently diagnosed differences in, e.g., ultraviolet (UV) flux to the troposphere (needed to initiate ozone photolysis for subsequent OH primary production) due to variations in cloud parameterizations and radiative transfer codes. Similarly, differences in the representations of volatile organic compound (VOC) oxidation pathways can influence the extent to which OH is recycled following reactions with hydrocarbons. Such nuances in the chemistry of OH make OH differences between models notoriously difficult to attribute. With properly coordinated simulations and sufficient model output, however, we have demonstrated that the barriers posed by complex, non-linear chemistry can be overcome.

The multi-dimensional system that describes OH behaviour is well-suited for study via machine learning approaches. We have previously demonstrated the utility of neural networks (NNs) for quantifying differences in OH among a small group of chemical transport models (CTMs), which rely on the specification of meteorological conditions (Nicely et al., 2017). Other groups have similarly shown the promise of machine learning techniques to better parameterize within models such complex processes as convection (Gentine et al., 2018), radiative transfer (Krasnopolsky et al., 2009), ozone production (Nowack et al., 2018) and deposition (Silva et al., 2019), and to replace the numerical integrators that simulate chemistry within models (Keller and Evans, 2019). NNs in particular are capable of modelling complex non-linear functions, making them a suitable technique for studying the non-linear chemistry involved in OH production and loss. The community continues to develop best practices for harnessing the power of machine learning for applications in atmospheric science. We build here on the specific application of NNs to better understand model representations of OH.

In this study, we apply an NN approach to quantifying the causes of OH differences to the large group of models that participated in the Chemistry-Climate Model Initiative (Eyring et al., 2013). We repeat our earlier analysis that identifies the primary drivers of OH and $\tau_{CH4}$ differences among model simulations conducted with specified dynamics, for a single year. We then expand the approach to study temporal variations in OH for 1980-2015, allowing for attribution of trends and interannual variability in $\tau_{CH4}$ to specific parameters. Finally, we compare the derived trends in OH simulated by the CCMI models to trends derived from a previous observation-based study.

## 2    Model Simulations

The Chemistry-Climate Model Initiative (CCMI), carried out as an official activity of the International Global Atmospheric Chemistry (IGAC) and the Stratosphere-troposphere Processes And their Role in Climate (SPARC) communities, seeks to enable inter-model evaluation of chemistry-climate models (Eyring et al., 2013). Phase 1 of CCMI has designed a set of simulations, covering both historical and future timeframes, with prescribed emissions inventories such that the interactive chemistry and its interplay with dynamical and radiative processes can be robustly compared between models. The analysis presented here focuses on one simulation, the historical specified dynamics (SD) simulation from 1980 to 2010 (REF-C1SD) (Hegglin and Lamarque, 2015; Morgenstern et al., 2017). Details of the emissions inventories recommended for this simulation can be found in Eyring et al. (2013). We have also performed the inter-model comparison portion of this analysis (Section 3.2) for the historical free-running simulation conducted from 1960 to 2010 (REF-C1). However, since a comprehensive examination of OH within the REF-C1 simulations was conducted by Zhao et al. (2019), those results are presented in the Supplement. We also include output from models that are not formal participants in CCMI, but provided simulations comparable to those being used here. These additional models are described below. Monthly mean fields are used for the various chemical, physical, and radiative parameters necessary for evaluating OH, described in Section 3. We analyse all models that include and provided output for the complete list of these variables.

Models that participated in the REF-C1SD simulation were nudged toward reanalysis meteorological fields such that dynamical conditions are represented with historical accuracy. The details of how nudging – of the winds, temperature, and sometimes pressure and water vapor fields – is conducted can be found in Morgenstern et al. (2017), Table S30. The nudging of these models to common fields does not necessarily improve model agreement, however, as in the case of large-scale tropospheric transport (Orbe et al., 2018). Particularly relevant to this analysis is the nudging of specific humidity, which is only performed in the MOCAGE model, of the models we analysed. Models that produced REF-C1SD simulations for CCMI and provided the necessary output to complete this analysis include: CAM4-Chem (Tilmes et al., 2016), EMAC-L47MA, EMAC-L90MA (Jöckel et al., 2016), MOCAGE (Josse et al., 2004; Guth et al., 2016), MRI-ESM1r1 (Deushi and Shibata, 2011; Yukimoto et al., 2012), and WACCM (Marsh et al., 2013; Solomon et al., 2015; Garcia et al., 2016). For both

configurations of the EMAC model, the simulations that included nudging of wave-0 temperatures were used (Jöckel et al., 2016). All models, here and including those described below, include interactive stratospheric chemistry.

Four models also contributed SD-type simulations to be analysed alongside the REF-C1SD CCMI simulations. The Goddard Earth Observing System (GEOS) model (Molod et al., 2015) conducted a "Replay" run, meaning the general circulation model computes its own meteorological fields for a 3 hour simulation period, then calculates the increment necessary to match a pre-existing reanalysis data set, in this case the Modern Era Retrospective Analysis for Research Applications version 2 (MERRA-2). The increment is then applied as a forcing to the meteorology at every time step during a second run of the same simulation

period. This simulation includes full interactive tropospheric and stratospheric chemistry from the Goddard Modeling Initiative (GMI) chemical mechanism (Nielsen et al., 2017) with output for years 1980-2018 at $0.625° \times 0.5°$ horizontal resolution and 72 vertical levels (Orbe et al., 2017; Stauffer et al., 2019; Wargan et al., 2018). This simulation is referred to as "GEOS Replay." Additionally, three chemical transport models (CTMs), which directly rely on established meteorological fields such as MERRA-2 rather than calculate them, provided output used in this analysis. The OsloCTM and GEOS-Chem

CTMs output all required variables for year 2000, while the GMI CTM (Strahan et al., 2013) simulated the full 1980-2015 period. All CTMs except GEOS-Chem calculate water vapor interactively in the troposphere. GEOS-Chem instead uses specific humidity fields from the MERRA reanalysis. We note that, while the GEOS Replay simulation described above used the GMI chemistry package, all discussion of the simulation from "GMI" refers to the separate, standalone CTM. While CTMs read in and use external meteorological fields rather than "nudging" or "replaying" internally calculated fields, we expect them

to similarly represent realistic meteorological conditions for a given year. As such, we group them with the REF-C1SD simulations from CCMI, bringing the total number of SD-type simulations analysed to ten.

## 3 Methods

### 3.1 Neural network setup

Neural networks are generated to predict the monthly mean OH mixing ratio for a given model following the method outlined

in Nicely et al. (2017). Briefly, four NNs are trained for one model, each for one simulation month. To reduce the computational demands of NN training, we only establish NNs for four months, one for each season: January, April, July, and October. Separate NNs are trained for the SD (main text) and free-running (Supplement) simulations, and all training is performed with output from year 2000. Each model gridbox located below the tropopause (thermal, following the WMO definition, for all models except GEOS Replay, which uses a "blended" tropopause calculation combining thermal and

potential vorticity definitions) is a single sample, so sample sizes are determined by a model's vertical and spatial native resolution. The number of tropospheric model grid points, and thus the training dataset sample size, is indicated for each model in Table S1 and always exceeds 100,000. Because separate NNs are trained for each month, and monthly mean output from each model simulation is used as input and training data, the dataset does not represent diurnal variations in OH chemistry.

The training process adjusts weighting factors such that mixing ratios of OH are predicted accurately when 3-D fields of the following variables are input to the NN: pressure, latitude, temperature (T), ozone ($O_3$), specific humidity ($H_2O$), methane ($CH_4$), the sum of nitrogen oxide and nitrogen dioxide ($NO_x \equiv NO+NO_2$), the ratio $NO:NO_x$, carbon monoxide (CO), isoprene ("ISOP"=$C_5H_8$), formaldehyde (HCHO), the photolysis frequency of $NO_2$ ("$JNO_2$"), the photolysis frequency of ozone to excited state $O(^1D)$ ("$JO^1D$"), and stratospheric ozone column ("$O_3$ COL"). Note that many of the inputs covary with one another depending on the chemical regime or meteorological conditions. A strength of the NN approach is that the inputs chosen need not be independent of each other. The $NO_x$ and $NO:NO_x$ inputs are calculated using monthly mean NO and $NO_2$ fields. All chemical species are input to the NN as unitless mixing ratios, except for methane, which is normalized by the maximum tropospheric value and indicated by the notation "$CH_4^{NORM}$". This normalization enables direct comparison of methane distributions between models, despite the fact that the use of boundary conditions sometimes results in substantially different amounts of methane between models. (While the CCMI models generally used roughly consistent boundary conditions, the additional simulations that were not formally part of CCMI exhibit methane concentrations outside the ranges of those in the CCMI models.) Pressure is provided in units of hPa, temperature in K, photolysis frequencies in $s^{-1}$, and $O_3$ COL in Dobson Units (DU). Three of the inputs – HCHO, $NO:NO_x$, and $O_3$ COL – have been introduced to this analysis since Nicely et al. (2017), due to availability of output from all models and to the added information they encompass that may be relevant for OH chemistry. For instance, having knowledge of the partitioning of $NO_x$ likely enables one to more accurately predict OH quantities compared to knowing just the total abundance of $NO_x$. Likewise, the introduction of $O_3$ COL is somewhat redundant when its primary effect on OH is through attenuation of ultraviolet (UV) flux to the troposphere, which is already encompassed by the input $JO^1D$. However, $JO^1D$ is also altered by other factors such as clouds, which cannot as easily be included as an input for this analysis (some models provide 2-D cloud fraction fields, others output 3-D fields, and still others do not give any metric regarding clouds). Whether strong differences in $JO^1D$ are caused by clouds or overhead ozone should be clarified by inclusion of $O_3$ COL as an input.

The neural network architecture is consistent with that of Nicely et al. (2017) and is shown in **Figure 1**. However, the number of computational nodes was doubled from 15 to 30 given the availability of more powerful computing resources. Two hidden layers each containing 30 nodes provided strong performance of the NN in reproducing the OH mixing ratios from a given model. For training, the model output is randomly split 80%/10%/10% into training, validation, and test datasets. During that process, the data from the training set is used to actively adjust weighting factors, and the validation set is evaluated to determine a training stopping point. When errors in predicting the validation data grow after adjusting weighting factors some number of iterations in a row, it is determined that the NN model prior to the growth in errors likely reached a local minimum in its cost function. This manner of "early stopping" helps to prevent over-fitting, though application of the NNs to alternative years is not immune to over-fitting, an issue discussed further in Section 4.3.1. For further application of this method across varying time scales, we would recommend a more methodical approach to sampling model output in time as well as in space. The final 10% of data is then used to independently test the resulting NN, and compare between different training iterations.

A total of five trainings were performed for each NN, and the NN with best performance (evaluated by the correlation coefficient from comparison of NN-calculated and model-simulated OH values) was chosen as the NN to be used in further analysis. Further details of the training process and evaluation metrics can be found in Nicely et al. (2017).

We note that alternative machine learning algorithms have seen increased application to problems within atmospheric science in the last few years, and may be equally or even better suited than neural networks to studying non-linear chemical systems. In particular random forest regressions and gradient boosting techniques offer greater computational efficiency and, in the case of random forests, have the capability to quickly identify which inputs are most strongly influencing the calculated output, known as "feature importance" (Hu et al., 2017; Grange et al., 2018; Liu et al., 2018; Keller and Evans, 2019). Additionally,
linear regression algorithms such as Ridge and Lasso regression may be beneficial in curbing issues related to extrapolation. We also do not intend to suggest that our chosen NN input list, architecture, and general method is the best approach; input variables were largely determined by available output, and architecture testing was conducted on the computing resources available at the time of the study. It is possible that a single NN could suffice for predicting OH variations throughout an entire year, rather than for just a single month, following methodical subsampling methods to create the initial training dataset. As
such, we encourage exploration of modifications to this method as well as additional algorithms for future machine learning applications to atmospheric chemistry.

## 3.2     Inter-model comparison approach

Once NNs are established for each model, an analysis is conducted to quantify the OH and $\tau_{CH4}$ differences attributable to individual input terms. To accomplish this, each model, *A*, is paired with another model, *B*, such that one input to the NN of
model *A* is substituted with the same field from model *B*. All other inputs are held fixed, using fields from model *A* for year 2000. Fields are interpolated to the resolution of the native model, *A* in this case, bilinearly across latitude and longitude, and linearly in log(pressure) space for the vertical coordinate. Any resulting changes in OH can then be directly attributed to the substituted variable.

The "swaps" that are performed in the manner described above undergo a process we refer to as "extrapolation control," which
restricts the substituted variable from leaving the range of values over which the native model's NN was trained. For example, if $O_3$ is being substituted from CAM4-Chem into the GMI NN, we not only check that a given CAM4-Chem $O_3$ value lies within the minimum and maximum GMI tropospheric $O_3$ values, but also that the GMI value of CO at that gridpoint can be associated with the new CAM4-Chem $O_3$ value. This check is performed across all variables, and essentially prevents the substitutions from venturing too far outside of the chemical regimes simulated within the native model. In the case that a
swapped variable exceeds the acceptable range of values, it is revised up or down accordingly. For reference, we tally the number of instances in which extrapolation control is invoked for two categories: coarse adjustments, when a NN input value from another model falls entirely outside the range of the NN input values from the native model, and fine adjustment, when

a value from another model must be tweaked to preserve the native model's chemical regimes. On average, coarse adjustments are incurred for 3.5% of all swapped data points, while fine adjustments are made to 18.8% of swapped values. We find that

extrapolation control is critical to achieve meaningful results with the NN inter-model comparison method, though it necessarily forces the attributed changes in OH and $\tau_{CH4}$ to be conservative estimates.

Metrics used to evaluate the results of variable swaps include tropospheric OH integrated columns for visualization and changes in $\tau_{CH4}$ for a globally-summed quantity. Tropospheric columns are integrated vertically and weighted by the mass of methane and the temperature-dependent rate constant of reaction between OH and methane. The global mean lifetime of

methane is found using Eq. 1:

$$\tau_{CH_4} = \frac{\sum M_{air} \times \chi CH_4}{\sum [OH] \times k_{OH+CH_4} \times M_{air} \times \chi CH_4},\tag{1}$$

where $M_{air}$ is the mass of air within a grid box, brackets denote number density, $\chi$ denotes mixing ratio, $k_{OH+CH4}$ is the reaction rate constant for the OH + CH$_4$ reaction calculated for each grid box temperature, and summations are performed over all tropospheric model grid boxes. This formulation is equivalent to the standard lifetime calculation of burden divided by loss

rate, adapted to the quantities most directly related to model outputs available (Chipperfield et al., 2014). Again, we note that this is strictly the atmospheric lifetime of methane with respect to loss by tropospheric OH. If one additionally includes all stratospheric grid boxes within the above summation, annual average lifetimes of almost all models consistently increase by ~1.2 years.

### 3.3      Time series evaluation approach

A new element of this analysis applies the already-established NNs of each model to examine the time evolution of OH over several decades of simulation. For this, we focus on the REF-C1SD simulation set, as it contains the most realistic representation of historical emissions and meteorological conditions, and thus is most likely to resemble true OH variations. All models that provided SD-type simulations as described in Sections 2.2 and 2.3 are included, with the exception of GEOS-Chem and OsloCTM, both of which only provided output for year 2000. Using a similar swapping technique as described in

Section 3.2, the NN for a given model is used to quantify the effect of substituting individual inputs from different years. No inter-model substitutions are conducted; instead, a single input is taken from the various years of the simulation (1980-2015) while all other inputs are fixed to their 2000 values. Because all swaps are performed on an intra-model basis, extrapolation control is largely unnecessary, since that model's chemical regimes do not vary drastically from the original year 2000 training output. However, we do see some instances, noted in Section 4.3, of anomalous behaviour in the $\tau_{CH4}$ results because some

variables undergo significant changes, particularly between the 1980s and training year 2000. Overall, the NN technique should be sufficiently generalizable to provide meaningful results even when using inputs lying modestly outside of the range

of training values. Robustness of the results is demonstrated by the emergence of several consistent features between the eight models examined, as discussed in Section 4.

## 4 Results and Discussion

### 4.1 Native model and NN performance

**Figure 2** shows values of $\tau_{CH4}$ found for all models that produced SD-type simulations. Annually and globally averaged lifetimes vary from 6.59 years (OsloCTM) to 8.41 years (GMI). All models exhibit the expected seasonal variation in $\tau_{CH4}$, with minimum values in the Northern Hemisphere (NH) summer months due to higher OH at this time of year. Specifically, the seasonal variation in the global mean is a result of greater anthropogenic influence in the NH and resulting increases in concentration of two OH precursors: ozone and $NO_x$.

An example of NN performance is shown for the January WACCM model in **Figure 3**, relative to the native model OH fields. Tropospheric OH columns are shown for the model and NN alongside the absolute value of the difference between the two. In general, the NNs from all models show similar magnitudes and spatial patterns in their calculated OH field, with errors somewhat randomly scattered and maximizing locally to values of ~10% of the total column value. Supplementary Figures S1-S4 show the performance of all NNs, for each of ten SD-type model simulations and for each of four months, while Table S2 provides further statistics on all NNs used here. Performance of all model NNs for year 2000 is strong, with values of $\tau_{CH4}$ calculated from the NN-generated OH field within 0.006 years of the parent model's $\tau_{CH4}$ on average. The maximum error in $\tau_{CH4}$, an overestimate by 0.012 years, occurs for the MRI-ESM1r1 model in the month of January. Performance is generally poorest in boreal winter, with average offsets in $\tau_{CH4}$ of 0.007 years, and strongest in boreal summer, for which the mean bias is only 0.004 years.

### 4.2 Inter-model comparison

The inter-model comparison component of this analysis can be understood fundamentally by the OH and $\tau_{CH4}$ differences generated by substituting input fields between models. An example of the OH column and $\tau_{CH4}$ changes that are calculated through individual variable swaps is shown in **Figure 4**. The two models with the highest and lowest values of $\tau_{CH4}$, GMI and OsloCTM, respectively, are chosen for this example. Swaps performed between the two models for the month of January reveal that local $O_3$, $JO^1D$, HCHO, and $NO_x$ account for the largest differences in $\tau_{CH4}$ for this particular model pairing. A complete budgeting of the changes in $\tau_{CH4}$ attributable to all inputs for GMI and OsloCTM is shown in **Table 1**. Note that the values of $\tau_{CH4}$ shown in Table 1 correspond to lifetimes for the month of January rather than annual averages and so will differ from the lifetimes noted at the beginning of Section 4.1.

It is worth discussing several features that are evident in the visualized OH changes shown in Fig. 4. First is the spatial distribution of the OH variations. Depending on how the sink or source term undergoing the swap affects OH chemistry, the strongest impacts may occur in localized areas or may distribute evenly over the globe. For instance, varying local ozone and $NO_x$ (Fig. 4a, b and 4g, h, respectively) exert the greatest influence on OH over the climatological tropics, with maximum impacts over land but extending over the oceans as well. This is likely a result of the anthropogenic or biomass burning emissions sources, which limit the largest differences in ozone and $NO_x$ between the two models to areas proximate to the South American, African, and Indonesian source regions for the month of January. The OH changes resulting from substitutions of the inputs $JO^1D$ and HCHO, however, are distributed over oceans as well as over land masses and, in the case of HCHO, are strongest in remote marine regions. This pattern is common for species that influence OH chemistry through mechanisms that are largely independent of local emissions. In the case of HCHO, its role as a secondary source of OH through methane oxidation is relatively more important in the absence of large VOC concentrations, thus its stronger influence is seen away from terrestrial vegetation.

The second feature to note in Fig. 4 is the symmetry between input swaps in opposing directions. In other words, the swap of an input from OsloCTM into the GMI NN generally yields OH column and $\tau_{CH4}$ changes that are equal but opposite to the changes resulting from use of a GMI input in the OsloCTM NN. With few exceptions, almost all regions of OH increase (red) in one model's NN are matched by OH decreases (blue) in the other model's NN in Fig. 4. The changes in $\tau_{CH4}$ are correspondingly similar in magnitude but opposite in sign. This behaviour is expected because a swap that may, e.g., increase an OH precursor and subsequently cause an increase in OH for one model will manifest as a decrease in that same precursor when the substitution occurs in the NN of the other model. While this pattern occurs for the vast majority of cases across all model pairings and swaps performed for this analysis, there are instances when symmetry is not maintained. This could happen for two reasons.

First, the sensitivities of the two models to a particular change in an OH precursor or sink could differ. For example, one model may be sensitive to an increase in isoprene, causing OH concentrations to drop in response. Another model may incorporate buffering effects, such as reactions involving oxidized volatile organic compounds (Taraborrelli et al., 2012; Lelieveld et al., 2016) that allow OH to be recycled following its reaction with isoprene, causing it to be less sensitive to the same change in methane. We refer to these variations in model sensitivities as chemical mechanism differences, as they are most likely a result of the chemical reactions, species representations, or reaction rates implemented within a model's chemical mechanism.

The second explanation for lack of symmetry in the OH response to a model swap is a forced asymmetry in the swapped inputs themselves, imposed by the extrapolation control technique described in Section 3.2. It is possible that the swap of an input in one direction, i.e. from Model *A* into Model *B*, could proceed with no alteration to the substituted variable, while the swap in the other direction, i.e. from *B* to *A*, results in the variable lying outside the trained range of Model A. The extrapolation

control process will revise the substitute variable field from Model $B$, such that the difference between it and the native field from Model $A$ is lessened. As such, the first swap into the NN of Model $B$ will yield a larger magnitude change in the input as compared to the swap into the NN of Model $A$. The impact of these factors is indirectly quantified through a remainder term that falls out of a full budgeting analysis, described below.

A third consideration in interpreting the information presented in Fig. 4 is the conditions that must be met in order for a large change in OH to manifest through this analysis. First, the two models between which a swap is conducted must exhibit differences in the parameter of interest. Should the two models exhibit, e.g., very similar ozone fields, then swapping one model's $O_3$ with the other's will produce little difference in the NN-calculated OH. Second, the model must have some OH sensitivity to the variable being swapped. If a model is insensitive to changes in methane, swapping in a drastically different methane field may not cause a perceivable difference in OH. Therefore, the absence of an OH response does not necessarily mean that input fields are similar between models. Conversely, the existence of large OH changes indicates that differences in the swapped input field exist between the two models *and* that the native model demonstrates a dependence of OH on that input variable.

A fourth issue is the fact that NNs can exhibit some degree of random behaviour, based on how they were trained and initialized. Our method involved training 5 NNs and selecting from those the one that performed best when compared to the independent test dataset. That single NN was used in all subsequent analysis. However, it is a useful exercise to evaluate the role of NN randomness in our results. We show in Figures S5 and S6 the left and right panels of Fig. 4, reproduced for the alternate NN trainings of the GMI and OsloCTM models, respectively. A visual comparison of tropospheric OH column differences among the five trainings of each model's NN reveals markedly similar spatial distributions and magnitudes. The values of calculated $\tau_{CH4}$ changes ($\Delta\tau_{CH4}$) do differ somewhat between the training instance, with larger effects on some variable swaps than for others. For instance, the standard deviation of the values of $\Delta\tau_{CH4}$ calculated for all five trainings of the GMI NN is about 0.2 years for the $J(O^1D)$ and HCHO swaps, but less than 0.05 years for $O_3$ and $NO_x$. We note, though, that some of the NNs displayed in Figures S5 and S6 exhibit worse performance than the one ultimately chosen for subsequent use. As a result of this exercise, the uncertainties resulting from this analysis method may be considered, at most, to be ~0.2 years.

The final point of interest in Fig. 4 is the general consistency in the signs of OH and $\Delta\tau_{CH4}$ for each model. The substitutions of all four variables generally cause an increase in OH within the GMI NN (and corresponding decrease in $\tau_{CH4}$) and a decrease in OH (increase in $\tau_{CH4}$) within the OsloCTM. This feature is most pronounced for this particular pair of models due to our reasoning for choosing them: they exhibit the largest difference in $\tau_{CH4}$ among our group of 10 models. Because the native GMI model has a longer $\tau_{CH4}$ value compared to OsloCTM, it makes sense that incorporation of OsloCTM's various OH precursor and sink fields into the GMI NN will tend to decrease the GMI $\tau_{CH4}$, bringing it into closer agreement with that of

OsloCTM. This characteristic points to the utility of this analysis as a budgeting tool for quantifying the cause of the difference in $\tau_{CH4}$ between two models. The $\tau_{CH4}$ accounting for the GMI and OsloCTM set of swaps conducted for January is shown in Table 1. When considering all 12 variable swaps that were performed, the NN analysis more than explains the original gap in $\tau_{CH4}$ between the two models. The GMI January lifetime of 9.24 years is decreased to 6.71 years ($\tau_{ORIG} + \Delta\tau$) after summing all $\Delta\tau$ values, while the OsloCTM lifetime is increased from 7.18 years to 9.48. This budgeting rarely provides a perfect accounting of the $\tau_{CH4}$ gap due to the same reasons that give rise to asymmetric OH responses to a given swap: chemical mechanism differences and asymmetric swaps of inputs due to extrapolation control. As a result, a remainder term, found as the difference between the other model's $\tau_{ORIG}$ and the present model's $\tau_{ORIG} + \Delta\tau$, is attributed to these factors. This term is listed in the last row of Table 1 with the label "Mech."

Results from analysing individual model pairs reveal a multitude of insights regarding idiosyncrasies in emissions of, global distributions of, and OH sensitivities to the various input parameters. These results, available in the archived data set described in Data Availability, may be especially useful to the reader with an interest in a particular species or model. However, with over 4000 plots (12 species × 10 models × 9 sub models × 4 months = 4320) and 180 $\tau_{CH4}$ budget tables generated, it is beyond the scope of this paper to highlight and explain every interesting feature. Instead, we aggregate the results across all models to identify some primary conclusions. **Figure 5** shows the change in $\tau_{CH4}$ for a specific model and substituted input variable, averaged over all nine pairings. For example, the data point shown for CAM4-Chem JO$^1$D is calculated from the nine $\Delta\tau_{CH4}$ values obtained when swapping the JO$^1$D fields from the other nine models into the CAM4-Chem NN. The circular point represents the mean of those nine values, while the whiskers indicate one standard deviation about the mean. Aggregate results shown in this manner are compiled both for individual months (available in the archived data set noted above) as well as for annually averaged output. The latter is calculated as the average of the four monthly mean and standard deviation values, and is shown in Fig. 5.

As with the individual OH tropospheric column change plots (Fig. 4), numerous conclusions can be drawn by studying the aggregated results in Fig. 5. The method for reading the data in Fig. 5 is demonstrated in the following example. The mean $\Delta\tau_{CH4}$ value attributable to JO$^1$D for the WACCM model is +0.99 years. This indicates that use of JO$^1$D fields from other models causes $\tau_{CH4}$ to increase by ~1 year, meaning the native JO$^1$D field from WACCM imparts a low bias to $\tau_{CH4}$ of 1 year, relative to the other models. A low $\tau_{CH4}$ would result from OH concentrations being too high. Since OH and JO$^1$D are positively correlated (i.e., JO$^1$D can be thought of as a source for OH) the too-high OH is an indication of too-high JO$^1$D. In general, positive values of $\Delta\tau_{CH4}$ correspond to relative high biases in input parameters that are source terms for OH and to low biases for species that instead serve as sinks. This reasoning is less straightforward for species such as HCHO, which can both produce and consume OH, while it is also produced by OH-initiated oxidation. We stress that these comparisons are strictly relative to other models, not to any observation or other indication of truth. So, points that appear as outliers in Fig. 5 should not necessarily be interpreted as an erroneous result, but rather should be considered as an area for further examination.

The ordering of variables along the x-axis of Fig. 5 denotes the average magnitude of $\Delta\tau_{CH4}$ values across all models, with parameters on the left accounting for the largest $\tau_{CH4}$ differences. As such, $JO^1D$ is the largest driver of OH differences in the CCMI SD model simulations, followed by local $O_3$ and $NO_x$. The subsequent variables $H_2O$, CO, the $NO:NO_x$ ratio, and HCHO cause moderate variations in tropospheric OH, while ISOP, $CH_4$, $JNO_2$, $O_3$ COL, and T are not responsible for inter-model spread in $\tau_{CH4}$. We note that T differences between the SD simulations are likely limited due the meteorological

constraints imposed on the models. However, examination of the free-running simulations, discussed in the Supplementary Material, also shows practically no impact of T on OH. Thus, we conclude that the effect of temperature on OH chemistry is likely indirect, acting through pathways embodied by other variables, such as $H_2O$ and species that exhibit strongly temperature-dependent reaction rates. Finally, the Mech. term, described in the discussion of Table 1, appears on the far right, indicating its origins as a remainder term from the budget analysis of individual model pairs. The magnitudes of $\Delta\tau_{CH4}$ values

attributed to chemical mechanism differences and asymmetric swaps between models are large enough to consistently rank the Mech. term third, between $O_3$ and $NO_x$, in terms of importance for OH in this analysis. Especially in model simulations conducted with common emissions inventories (though inventories can be implemented very differently among models, as demonstrated by Young et al. (2013)), we expect some of the disparity in a short-lived species like OH to emerge from differences in chemical mechanism implementations. In other words, when responses in OH to a given change in a source or

sink term differ between two models, the remainder term (or term labelled "Mech." in Table 1) will increase, representing variations in the sensitivity of OH that presumably arise due to the two different implementations of the chemical mechanism. It is possible that other factors are represented by this term; e.g., other chemical species that influence OH chemistry but are not considered in the NN analysis could contribute to the Mech. term. However, previous analysis using a 0-D chemical box model as a "standard" mechanism in Nicely et al. (2017) suggested a correlation between actual biases in OH imparted by a

given model's chemical mechanism and the remainder term resulting from the NN analysis. Therefore, we have some confidence that the Mech. term is meaningful, though significant further study would be required to parse the actual mechanistic differences responsible for imparting bias in OH calculations.

Significant inter-model differences in the largest driver of $\tau_{CH4}$ spread, $JO^1D$, could arise from two possible sources. The amount of solar UV light penetrating down to the troposphere is largely dictated by the stratospheric column ozone amount.

However, the differences in total ozone column are generally small and insufficient to cause the variations in $JO^1D$ seen among the CCMI models. Rather, $JO^1D$ likely varies to a great extent due to differences in cloud cover, and dissimilar treatments of clouds within model photolysis codes. Figure S7 highlights this effect by showing the ratio of $JO^1D$ at the surface to $JO^1D$ in the upper troposphere (UT) for each model. The relatively small column amounts of ozone within the troposphere should account for very little absorbed UV light, making it much more likely that deviations in this ratio from 1.0 are driven by

scattering due to clouds and possibly aerosols. The fact that models show large spatial differences in this ratio is a strong indication that clouds underlie the model differences in $JO^1D$.

While the model differences in JO$^1$D, O$_3$, NO$_x$, and chemical mechanisms appear to drive the bulk of the $\tau_{CH4}$ spread among this group of CCMI models, we emphasize that individual models may not adhere to these conclusions. As such, any efforts to improve a particular model should instead focus on the results specific to that model. For instance, HCHO plays a very

small role in describing inter-model differences in OH on average, but for the OsloCTM model, HCHO is a much more important factor. Thus, we refrain from offering an across-the-board solution for remedying the large model spread in $\tau_{CH4}$ and instead suggest a more individualized approach of studying plots such as those shown in Fig. 4 for more spatially and temporally resolved information. Visualizations of all model swaps, for all months and species, are available in our archived data set described in Data Availability for this purpose.

There are several other qualifications to note when considering the results of the inter-model comparison. One is the negating effect between the JO$^1$D and tropospheric O$_3$ variables. Many, but not all, model $\Delta\tau_{CH4}$ values for JO$^1$D in Fig. 5 are opposite in sign to the $\Delta\tau_{CH4}$ values attributed to O$_3$. Physically, photolysis of tropospheric ozone by light at wavelengths below 336 nm to form excited state O($^1$D) and subsequent reaction with H$_2$O to form OH is a loss pathway for ozone. Therefore, more UV flux will tend to decrease tropospheric ozone concentrations while increasing OH, and vice versa. This physical

mechanism, then, can explain the frequent cancellation of the $\Delta\tau_{CH4}$ values attributed to these two factors. Should a modeler attempt to alter a model's OH field by forcing adjustments in its JO$^1$D, the opposing impact of tropospheric O$_3$ may result in no change for the value of $\tau_{CH4}$. However, this does not preclude the finding that both JO$^1$D and tropospheric O$_3$ are substantially different in the models for reasons we do not fully understand. Tropospheric ozone can also vary between models for reasons external to the radiative environment. For instance, differences in the stratosphere-troposphere exchange, wet and

dry deposition, and lightning NO$_x$ emissions can each cause substantial variations in tropospheric ozone among models (Wild, 2007). Further parsing of the reasons for the ozone differences seen among the CCMI models is difficult without specialized output, including tracers such as ozone of stratospheric origin and NO$_x$ generated by lightning. We recommend a targeted study to address the underlying reasons for the variations in tropospheric ozone.

Another qualification concerns the issue of causation versus correlation. Machine learning techniques, and NNs in particular,

are generally more adept at identifying the predictors of a certain phenomenon than traditional methods, such as multiple linear regression. However, it is still possible that an input that is tightly correlated with the output may be misidentified as a driver of variations in the output. This is particularly relevant to keep in mind for species that serve as sinks of OH, such as CO and methane. Whether a decline in OH initiates or results from an increase in its sinks is difficult to differentiate, even with advanced analysis methods. Therefore, descriptions of CO and methane as drivers of OH variations in this text may just as

well be interpreted conversely, as downstream indicators of the change in oxidizing capacity.

A final qualification is this analysis constitutes a foundationally hypothetical experiment. It essentially addresses the questions, "What if we could switch the fields of just one chemical species between two global models? What would be the instantaneous

impact on OH? on $\tau_{CH4}$?" This approach, then, necessarily neglects the roles of feedbacks in the atmospheric system (e.g., if the $NO_x$ field is perturbed, this will propagate to changes in ozone as well, with time). However, for the objective of teasing apart the influences on global OH abundance and $\tau_{CH4}$ and explaining inter-model differences, a notoriously difficult task, we regard our approach as a valuable exercise.

## 4.3    Time series evaluation

The second half of our NN analysis interrogates temporal trends in OH and $\tau_{CH4}$. **Figure 6** shows the evolution of $\tau_{CH4}$ in the SD-type simulations conducted for 1980-2010. Two models, GEOS-Chem and OsloCTM, only provided output for year 2000, and so only appear as single points in Fig. 6. In addition, some models provided output beyond year 2010; output from years through the end of 2015 was included when available. The lifetimes all show a general downward trend over time, consistent with the upward trend in global mean tropospheric OH concentration shown by Zhao et al. (2019b) (their figure 4). Results concerning attribution of the $\tau_{CH4}$ time series are presented in subsection 4.3.1, while derivation and analysis of trends are shown in subsection 4.3.2.

### 4.3.1    Attribution of the $\tau_{CH4}$ time series

Swaps of input variables to a NN are conducted on an intra-model basis, with the goal of determining which OH precursors and sinks are responsible for OH variations over time. The results of these swaps are shown for each model in **Figure 7**. Changes in $\tau_{CH4}$ attributable to each parameter are displayed as a function of year. Because we use the same NNs established for the inter-model comparison described in Section 3.2 trained on output from year 2000, the values of $\Delta\tau_{CH4}$ for all species in year 2000 of Fig. 7 is zero by design. As an input field from another year is swapped into the NN, however, OH differences manifest and are denoted by the corresponding change in $\tau_{CH4}$. Because we are relying on the same NNs used for the inter-model analysis, we emphasize that the methane fields used here are still normalized, separately for each year. As a result, the variations in $\tau_{CH4}$ due to $CH_4^{NORM}$ should not be interpreted as a measure of the methane feedback factor (Prather et al., 2001; Fiore et al., 2009; Holmes et al., 2013; Holmes, 2018). Instead of representing the change in OH with a change in absolute concentration of methane, the numbers shown here signify the change in OH with a change in how methane is distributed within the atmosphere, both vertically and spatially. Largely, one would expect this to remain constant over time, though results from this analysis of the CCMI simulations suggests there are some modest changes in $\tau_{CH4}$ attributed to the distribution of tropospheric methane. Should a similar method be applied to analysis of temporal variations in OH in the future, we would encourage training the machine learning algorithm on data spanning all years such that use of methane absolute values would be possible.

While significant diversity in the drivers of OH variability across models is evident from Fig. 7, there are also several distinctive features that appear repeatedly. For instance the response of $\tau_{CH4}$ to changes in CO shows a prominent peak in year

1998 in all models except one. To gauge the role of emissions in this response, we show in Supplemental Figures S8-12 the time series of CO mixing ratios and other parameters averaged for the region most impactful to $\tau_{CH4}$: the tropical lower troposphere (latitudes between 30°S and 30°N, pressures greater than or equal to 700 hPa). Indeed, CO mixing ratios maximize in almost all models in year 1998, likely as a result of the emissions inventory reflecting the extreme biomass burning and strong El Niño Southern Oscillation (ENSO) event during that and the preceding year (Duncan et al., 2003 and references therein). The increase in $\tau_{CH4}$ can thus be explained by the increased CO sink of OH, causing a temporary depletion of the oxidant. In addition, less distinctive peaks in $\tau_{CH4}$ due to CO are identified in other years with strong El Niño conditions, notably 1982-1983, 1987, and 1991-1992 (Duncan et al., 2003).

The impacts of several other variables on $\tau_{CH4}$ also demonstrate behaviour with reasonably identifiable causes. A prolonged decrease in $\tau_{CH4}$ due to $JO^1D$ from 1992 to 1998 is evident in the analysis of the CAM4-Chem, GEOS Replay, GMI, MRI-ESM1r1, and WACCM NNs. This may correspond to several confounding events that acted to increase the flux of UV light to the troposphere, increasing the primary production of OH and decreasing $\tau_{CH4}$, as seen in Fig. 7. First, solar activity reached a maximum around 1990, after which the decline in sunspots correlated strongly with a decline in tropical total ozone columns (Duncan and Logan, 2008). Second, the eruption of Mount Pinatubo in 1991 likely impacted $JO^1D$ through the decrease in stratospheric ozone that resulted (Tie and Brasseur, 1995; Aquila et al., 2013). Finally, the prolonged ENSO event of 1990-1995 (Allan & D'arrigo, 1999) may have caused reduction in cloud cover due to drought conditions (Duncan et al., 2003). Interestingly, the $\tau_{CH4}$ response to $H_2O$ is moderately anticorrelated with CO. This is particularly evident for year 1998 in many of the models, when large biomass burning events occurred in many regions of the world, such as the boreal forests of both Asia and North America, Central America and Mexico, and Indonesia, which were attributed in part to a strong El Niño in 1997 that transitioned in a strong La Niña in 1998. Although strong ENSO events cause drought conditions over some regions, it is more fundamentally associated with warming sea surface temperatures and increased evaporation, particularly in the tropical Pacific Ocean. Thus, it is reasonable that larger values of specific humidity will tend to increase OH primary production during an El Niño year, as suggested by the decrease in $\tau_{CH4}$ shown in Fig. 7. An apparent increase in ozone also coincides with the 1998 ENSO event, determined by the decreasing component of $\tau_{CH4}$. The prevalence of biomass burning would indeed cause increases in tropospheric ozone through increased emissions of its precursors, CO, VOCs, and $NO_x$. Additionally, the $\tau_{CH4}$ response to $O_3$ shows the most distinguishable trend of all the variables over the full 1980-2015 period. Steady decreases in $\tau_{CH4}$ due to $O_3$ imply an increasing tropospheric ozone burden, a modelling result supported by observations (Verstraeten et al., 2015).

We also note the appearance of spurious results in several cases. The $\tau_{CH4}$ responses to $CH_4^{NORM}$ in EMAC-L47MA and EMAC-L90MA as well as to $O_3$ COL in MOCAGE extend to very large negative values in the early part of the time series. To show the full extent of the EMAC $\tau_{CH4}$ responses to $CH_4^{NORM}$, we show alternate versions of Figs. 7b and 7c with expanded

y-axis ranges in Supplementary Figure S13. Chemical conditions during the 1980s would differ most markedly from the regimes simulated in year 2000, on which the NNs are based. Particularly for concentrations of methane, which underwent monotonic rise aside from a stabilisation period from 2000 to 2007 (Turner et al., 2019), conditions in 1980 could be quite different. However, as was noted in Section 3.1, methane inputs to the NNs are normalized against the maximum tropospheric value. The field of $CH_4^{NORM}$ for each year is likewise normalized against the maximum methane for that year, so a strong response in $\tau_{CH4}$ must indicate a significant change in the distribution of methane, not just in changes in its concentration over time. Indeed, Supplementary Figure S14 shows the normalized methane values used as input to the NNs for the pressure level closest to the surface. For each EMAC configuration (for the month in which the $\tau_{CH4}$ response shown in Fig. 7 is largest and most unphysical), the methane distributions in the 1980s do show notable change from the year 2000 distribution used for training. Specifically, methane mixing ratios in the Southern Hemisphere drop relative to the larger concentrations in the Northern Hemisphere. Other models, such as WACCM shown in the bottom panels of Fig. S14, show practically no inter-annual change in the methane distribution for a given month. This behaviour in the EMAC model likely results from implementation of a Newtonian relaxation scheme to determine a time-varying, latitude-dependent lower boundary condition for methane (Jöckel et al., 2016). Our spurious NN result may indeed be explained by a slowdown in the rate of increase in methane concentrations at the lower boundary initiated in 1980, evident in supplementary figure E1 of Jöckel et al. (2016). While this method of determining boundary conditions generally represents a more sophisticated treatment of methane, within the context of this analysis, it imparts an artificially strong signal in OH and $\tau_{CH4}$. Therefore, the unphysical results in Fig. 7b and 7c due to $CH_4^{NORM}$ indicate an artefact due to the NN method, not a problem in the EMAC model itself.

For the other occurrence of anomalous behaviour, MOCAGE shows an unrealistically large response of $\tau_{CH4}$ to $O_3$ COL in the 1980s (Fig. 7f), a result not corroborated by any other model. Supplementary Figure S15 illustrates the likely cause of this behaviour. While most models exhibit modest changes in total $O_3$ COL between 1980 and 2000, including GEOS Replay shown in the top set of panels, the MOCAGE model (bottom panels) shows much larger column amounts in year 1980. These values fall well outside the range of $O_3$ COL amounts on which the NN was trained, so unrealistic behaviour of the NN in this case is not surprising.

These examples of spurious results highlight an issue that must be treated with caution when using machine learning approaches. Because the application of our NN method to time series analysis is an extension beyond the originally intended purpose, not all NNs are sufficiently generalizable to reliably reproduce OH for years other than the training year, 2000. To account for this, we evaluate each NN for all years by inputting variables from each year. With this test, all inputs are changed, not just a single input at a time. The resulting OH, as depicted in Figures S16-S23 for select years, compares well to the native model's OH field for that year in many cases, but not in all. Considerable bias occurs at low OH mixing ratios, though we note that near-zero concentrations will likely not affect the resulting globally-integrated $\tau_{CH4}$ unless values are grossly overestimated. This evaluation also represents a rigorous test of the NNs, as significant shifts in numerous inputs at once

might push the NN algorithm into new phase space not encountered during training, much more so than only changing one input at a time, which is our approach in the subsequent time-series analysis. Nonetheless, we limit the influence of poorly generalizable, or "overfit," NNs by only including in the multi-model mean results for the years in which a NN reproduces its native model's OH field with an $r^2$ value greater than or equal to 0.95. For four NNs (one per month) created for each of 8

CCMI models, across 36 years, the potential application of the NNs to 1152 calculations (4×8×36) is reduced to 696 calculations using this test. Results from this point forward are subject to this quality check, and were found to be insensitive to the $r^2$ threshold imposed. This insensitivity is demonstrated by alternate versions of the figures to come, placed in Supplement, generated using all NNs rather than the quality-filtered NNs.

**Figure 8** shows the multi-model mean attribution of variations in $\tau_{CH4}$. Many of the same features identified in Fig. 7 also

emerge here: clear definition of strong ENSO years in the CO response, apparent Mt. Pinatubo effects in the $JO^1D$ response, and a general downward trend in $\tau_{CH4}$ due to $O_3$ are all observed. Also, as might be expected from the inter-model comparison results discussed in the prior section, $JO^1D$, $O_3$, $NO_x$, $H_2O$, and CO account for many of the strongest OH variations over time (Fig. 7) as well as between models (Fig. 5). Supplementary Figure S24 shows the analogue of Fig. 8, without the quality filter applied to the NNs described above. I.e., all NN results from Fig. 7 are included, except the spurious cases of EMAC $CH_4^{NORM}$

and MOCAGE $O_3$ COL.

#### 4.3.2    Trends and interannual variability in the $\tau_{CH4}$ time series

We also perform linear fits to each response time series in Fig. 8. The resulting trends in $\tau_{CH4}$ are shown in **Figure 9**, panel (a). The interannual variability of $\tau_{CH4}$ is also calculated as the standard deviation of the detrended time series, shown in Fig. 9b, though it is relevant to note that CTMs have historically not captured the full interannual variability exhibited by observed

OH proxies (Holmes et al., 2013). Supplementary Figure S25 shows the equivalent of Fig. 9, without application of the NN quality filter described above. Negative trends in $\tau_{CH4}$ due to $O_3$, $H_2O$, $JO^1D$, and $NO_x$ stand out as largest in magnitude. The sum of all factors shown in Fig. 9a is –2.3±0.4% decade$^{-1}$, which is comparable to the mean downward trend in $\tau_{CH4}$ seen in Fig. 6, –1.8% decade$^{-1}$. Time series of the model input variable fields show corresponding trends, with parameters that serve as source terms of OH increasing over time (Supplemental Figures S9-12). Tropospheric ozone and $NO_x$ show clear upward

trends over time, while $H_2O$ and $JO^1D$ show upward trends with more variability, which is also conveyed by the error bars in Fig. 9a. It is interesting to note that $H_2O$ plays a stronger role in the overall temporal trend of $\tau_{CH4}$, as compared to its role in explaining inter-model differences. This is likely due to the fact that temperatures were constrained in the specified dynamics simulations, which in turn should determine the water vapor calculated within the models. The interannual variability attributed to CO in Fig. 9b is also consistent with the large year-to-year swings in tropical lower tropospheric CO mixing ratios

shown in Supplemental Figure S8. While Fig. 9a suggests that CO exhibits very little overall trend between 1980 and 2015,

we note there is a discernible increase in CO prior to ~1998 in Fig. S8 followed by a steady decline thereafter. This is consistent with remote site measurements that show significant negative trends in CO since the late 1990s (Zeng et al., 2012).

Finally, the attributed trends in $\tau_{CH4}$ from the CCMI models (Fig. 9a) are compared in **Figure 10** to trends in tropospheric mean OH concentration ("$[OH]^{TROP}$") from a previous observation-based analysis (Nicely et al., 2018). In that work,
TOMS/OMI/SBUV observations of total column ozone were used to infer radiative effects on the OH burden, while water vapor from the AIRS instrument, methane from surface observations, $NO_x$ from a global model simulation constrained to realistic emissions, and temperature from the MERRA-2 reanalysis were analysed to calculate chemical impacts on $[OH]^{TROP}$. In Nicely et al. (2018), the trend in $[OH]^{TROP}$ due to $NO_x$ encompassed the effects of both the total abundance and the partitioning of $NO_x$, while the $O_3$ COL factor encompassed all radiative effects on OH. Thus, to perform a "like for like"
comparison, the $\tau_{CH4}$ trends due to $NO_x$ and $NO:NO_x$ are combined, as are the trends due to $O_3$ COL and $JO^1D$ shown in Fig. 9a. Error bars shown in Fig. 10 represent the $1\sigma$ uncertainty in the slope of the linear fit and, in the case of combined trends, are found by summing in quadrature the individual uncertainties. Because $\tau_{CH4}$ varies with the inverse of OH concentration, note that the x-axis of Fig. 10 is inverted and a $-1{:}1$ line is shown in grey.

The trends in $\tau_{CH4}$ from this analysis and in $[OH]^{TROP}$ from Nicely et al. (2018) are in reasonably good agreement for $H_2O$,
$NO_x$, $O_3$ COL, and temperature. In particular, the two trends due to $H_2O$ agree within the uncertainties, with $\tau_{CH4}$ decreasing by ~0.5 % decade$^{-1}$ and $[OH]^{TROP}$ increasing at almost the same rate. The impacts of $NO_x$ and $O_3$ COL are found to increase OH concentrations in both studies, though the impacts on $\tau_{CH4}$ from the CCMI models are found to be larger in magnitude than the observational estimate. The small impact of temperature, tending to lessen the OH burden, is also in close agreement between the two studies, with the CCMI models again showing a slightly stronger response. The role of $NO_x$ in driving ~0.3
% decade$^{-1}$ decline in $\tau_{CH4}$ is roughly consistent as well. Only the effect of ozone column falls relatively far from the $-1{:}1$ line, with analysis of the CCMI models suggesting a stronger decrease in $\tau_{CH4}$ between 1980 and 2015, albeit with large uncertainties. This may result from inaccurate representations of stratospheric ozone in the CCMI models, mischaracterization of the impacts on UV photolysis in the troposphere, or a combination of both. Overall, the results depicted in Fig. 10 show relatively robust findings regarding the responses of $[OH]^{TROP}$ and $\tau_{CH4}$ to the factors examined through these two independent
studies.

Because the methane used as input for the CCMI NNs was normalized, as discussed above, the trend in $\tau_{CH4}$ found in this analysis due to $CH_4^{NORM}$ did not represent a methane feedback factor in the traditional sense. As such, it is not comparable to the trend in $[OH]^{TROP}$ due to methane found by Nicely et al. (2018) and so was not included in Figure 10. However, even in the event that one were to retrain new NNs using absolute values of methane and sampling across all years to generate the
training dataset, we would question the physical meaning of the resulting trends. With the current necessity of providing boundary conditions for surface methane rather than fluxes in models, our ability to realistically simulate methane is hampered.

We encourage the further examination of the response of OH to methane on the global scale, which is likely a large influencer of tropospheric OH abundance, as indicated in Nicely et al. (2018) and Holmes et al. (2013).

## 5    Conclusions

We perform a neural network analysis of the monthly mean output from historical simulations of ten models that participated in CCMI for the purposes of understanding OH and $\tau_{CH4}$ differences and temporal trends.  NNs are trained to reproduce OH mixing ratios for a given model using 3-D fields of 12 OH precursor and sink parameters.  Performing swaps of the NN inputs between models produces a quantitative estimate of the difference in $\tau_{CH4}$ that can be attributed to variations in the substituted variable.  Among the ten models that we examine, on average, variations in $JO^1D$, local $O_3$, $NO_x$, and chemical mechanisms

account for the largest differences in $\tau_{CH4}$.  Model diversity in representations of $H_2O$, CO, the partitioning of $NO_x$, and HCHO is responsible for moderate OH differences, while isoprene, $CH_4^{NORM}$, $JNO_2$, overhead ozone column, and temperature account for little-to-no variation in OH.  However, the relative importance of a particular variable is highly model-dependent, so any effort to improve the representation of OH within a given model should be guided by that particular model's results.

We also analyse time series of $\tau_{CH4}$ using the year 2000 NNs generated for the first half of the study.  All models exhibit a

downward trend in $\tau_{CH4}$ between 1980 and 2015, varying from $-0.54$ % decade$^{-1}$ to $-2.97$ % decade$^{-1}$ (average of $-1.83$ % decade$^{-1}$).  Swaps of NN inputs are conducted between years rather than between models, so attributions of the factors influencing trends in $\tau_{CH4}$ are found for each model and then combined into a multi-model mean result.  This analysis indicates that the largest contributors to the decreasing trend in $\tau_{CH4}$ are $O_3$, $JO^1D$, $NO_x$, and $H_2O$, while CO also imparts a large degree of interannual variability.  Features due to strong ENSO events and associated biomass burning as well as the eruption of

Mount Pinatubo are discernible in the time series of attributed variations in $\tau_{CH4}$.  In particular, the species CO, $H_2O$, and $O_3$ instigate prominent responses during strong El Niño years.  Finally, the attributed trends in $\tau_{CH4}$ from the NN analysis of CCMI model output are compared to trends in tropospheric mean OH concentration found previously in the observation-based study of Nicely et al. (2018).  While the strong response of $\tau_{CH4}$ to increasing $H_2O$ over time appears to be a robust result, disagreement on the methane feedback on OH between the two studies highlights limitations in the approaches of both, in

addition to more systemic issues in the community's ability to model methane.

The NN and machine learning methods in general provide a valuable tool for performing insightful model intercomparisons of complex systems in a computationally-efficient manner.  These approaches, however, must be undertaken with care to avoid erroneous results and recognition of their limitations.  At present, we have devised a method to identify the drivers of OH variations, whether between models or between years, at coarse temporal resolution.  Much future work is needed, though:

observations must be incorporated to introduce a ground truth element to this analysis in a manner that either adjusts for or avoids  disconnects  between  coarse  versus  local/instantaneous  spatiotemporal  scales  and  appropriately  accounts  for

measurement uncertainty; an analysis of model output with much higher temporal frequency is needed to identify exactly where model differences in chemical mechanisms lie; and subsequent studies of why the various OH precursor and sink fields differ are required to make this analysis of greatest utility for improving model representations of $\tau_{CH4}$. While these challenges

are significant, they are not insurmountable, especially as machine learning and other advanced statistical analysis techniques continue to be developed and honed.

**Data Availability**

All output from most of the models that participated in CCMI is available at the Centre for Environmental Data Analysis (CEDA), the Natural Environment Research Council's Data Repository for Atmospheric Science and Earth Observation, at

http://data.ceda.ac.uk/badc/wcrp-ccmi/data/CCMI-1/output. WACCM and CAM4-Chem output for CCMI is available for download at http://www.earthsystemgrid.org. For instructions for access to both archives see http://blogs.reading.ac.uk/ccmi/badc-data-access. Output from the models that were not formal participants in CCMI Phase 1 is available from the co-authors who performed the model simulations; please contact the corresponding author with requests. A complete set of figures and tables generated by the model intercomparison and time series analyses is available at

https://doi.org/10.13016/vvbp-p6o8 (Nicely et al., 2020).

**Acknowledgments**

JMN was supported by an appointment to the NASA Postdoctoral Program at the NASA Goddard Space Flight Center, administered by the Universities Space Research Association under contract with NASA. The authors also acknowledge the joint WCRP SPARC–IGAC Chemistry-Climate Model Initiative (CCMI) for organizing and making available the suite of

model simulations used here. Special thanks is extended to the modeling groups which, at times, provided extra output that enabled this inter-comparison to take place with the maximum number of participants. We also thank many colleagues who engaged in helpful discussions that shaped the direction of this work and inspired additional analyses, including, but not limited to: Vaishali Naik, Sarah Strode, Jason St. Clair, and Melanie Follette-Cook. This work was partly supported by JSPS KAKENHI Grant Number JP19K12312. LER acknowledges China Southern for partial support. The EMAC simulations

have been performed at the German Climate Computing Centre (DKRZ) through support from the Bundesministerium für Bildung und Forschung (BMBF). DKRZ and its scientific steering committee are gratefully acknowledged for providing the HPC and data archiving resources for this consortial project ESCiMo (Earth System Chemistry integrated Modelling). GZ and OM acknowledge funding under the New Zealand Government's Strategic Science Investment Fund (SSIF).

## Author contributions

JMN and RJS conducted initial design of the method. JMN carried out the analysis. Development and refinement of the analysis were further guided by BND, GMW, and TFH. All other authors provided model output central to the analysis. JMN drafted the manuscript, and all co-authors provided assistance in finalizing the figures and text.

## Competing interests

The authors declare that they have no conflicts of interest.

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

**Table 1. Accounting of CH<sub>4</sub> lifetime differences between GMI and OsloCTM simulations for January, 2000.**

| | | GMI | OsloCTM |
|---|---|---|---|
| $\tau_{CH4,ORIG}^{a}$ (year) | | 9.24 | 7.18 |
| $\Delta\tau_{CH4}$ due to[b]: | $O_3$ | –0.91 | +0.79 |
| | $JO^1D$ | –0.59 | +0.60 |
| | HCHO | –0.64 | +0.51 |
| | $NO_x$ | –0.45 | +0.33 |
| | $JNO_2$ | –0.34 | +0.15 |
| | Isoprene | –0.03 | +0.28 |
| | CO | +0.19 | –0.07 |
| | $H_2O$ | +0.10 | –0.13 |
| | $CH_4^{NORM}$ | +0.11 | –0.06 |
| | $NO/NO_x$ | +0.07 | –0.05 |
| | $O_3$ COL | –0.02 | –0.06 |
| | T | –0.02 | +0.00 |
| $\Delta\tau_{CH4,TOT}^{c}$ | | –2.52 | +2.30 |
| $\tau_{CH4,ORIG} + \Delta\tau_{CH4,TOT}$ | | 6.71 | 9.48 |
| Mech.[d] | | +0.47 | –0.24 |

[a]$\tau_{CH4,ORIG}$ represents value of $\tau_{CH4}$ evaluated directly from the model.

[b]$\Delta\tau_{CH4}$ calculated from output of NN when noted variable is substituted with values from the other model.

[c]Sum of all $\Delta\tau_{CH4}$ values calculated for each input substitution.

[d]Remainder of original $\tau_{CH4}$ difference not accounted for by NN substitutions; calculated as $\tau_{CH4,ORIG}$ (model A) – [$\tau_{CH4,ORIG}$ (model B) + $\Delta\tau_{CH4,TOT}$ (model B)].

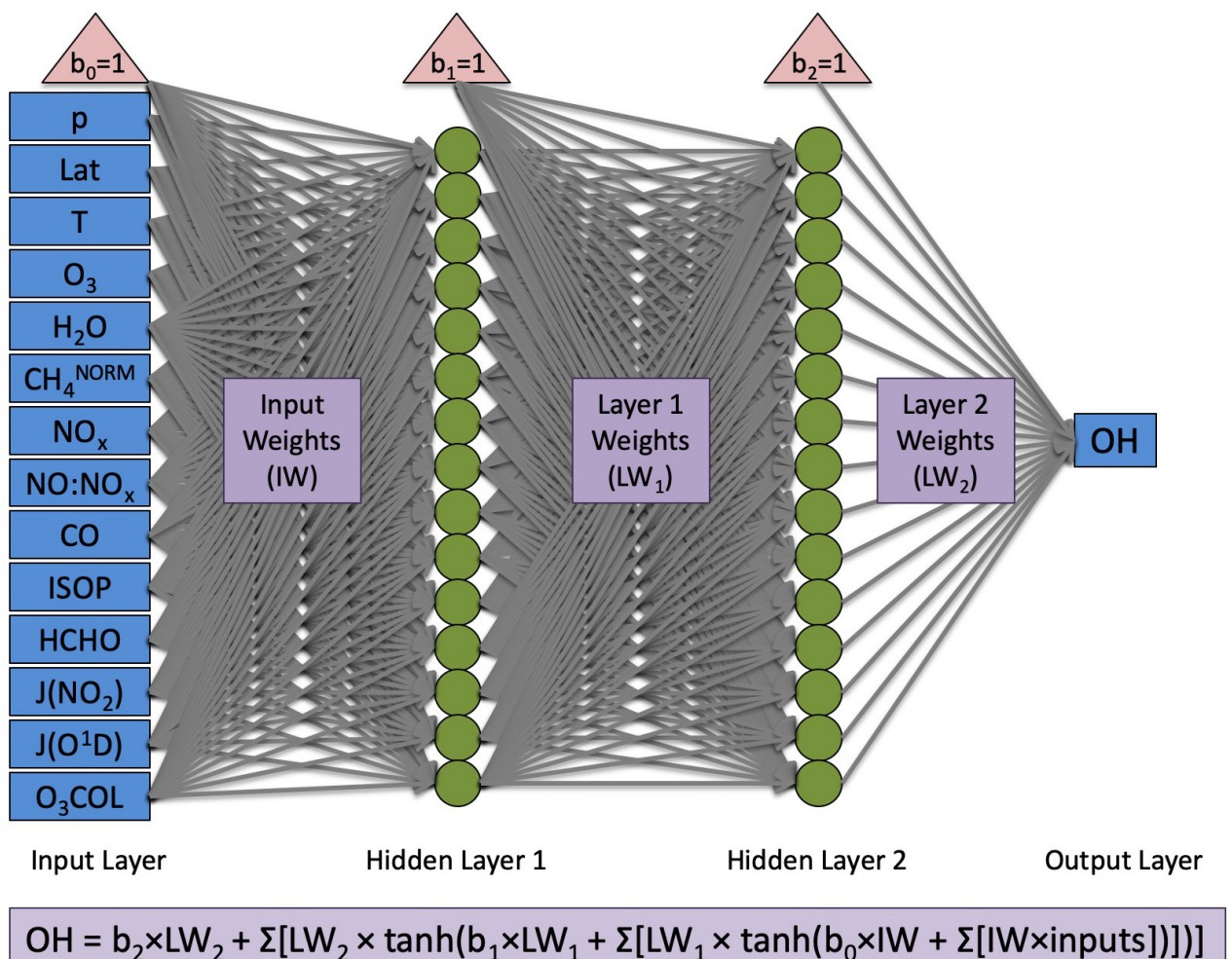

$$OH = b_2 \times LW_2 + \Sigma[LW_2 \times \tanh(b_1 \times LW_1 + \Sigma[LW_1 \times \tanh(b_0 \times IW + \Sigma[IW \times inputs])])]$$

**Figure 1. Architecture for neural networks generated in this study. Blue boxes designate inputs (left) and output (right), red triangles indicate bias terms, green circles indicate nodes at which activation functions are performed, and grey arrows represent the weighting terms, which are optimized through the training process. For full details of the neural network setup and training, we refer readers to Nicely et al. [2017]. Although 15 nodes are shown here in each hidden layer, 30 were actually used for all NNs in this study.**

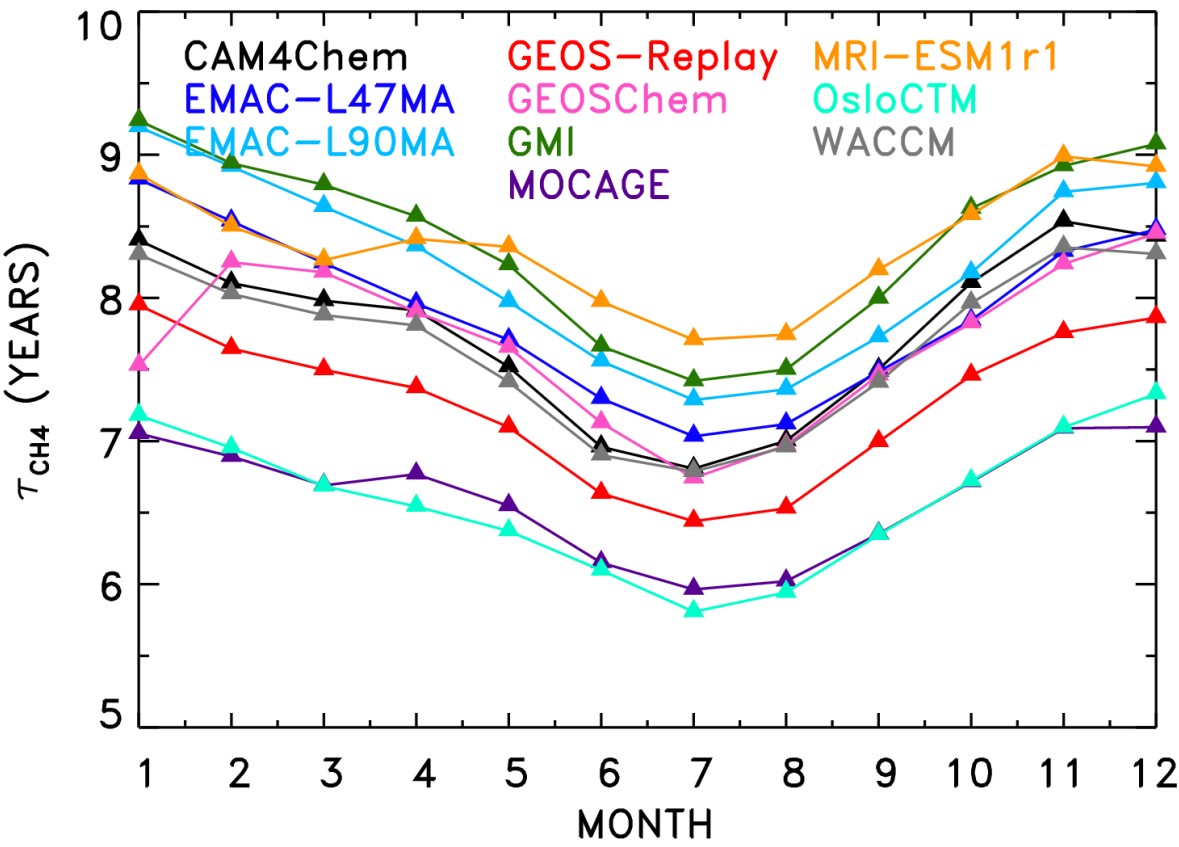

**Figure 2.** Seasonal variation in CH$_4$ lifetime for year 2000 for the CCMI specified dynamics (REF-C1SD) and chemical transport model simulations.

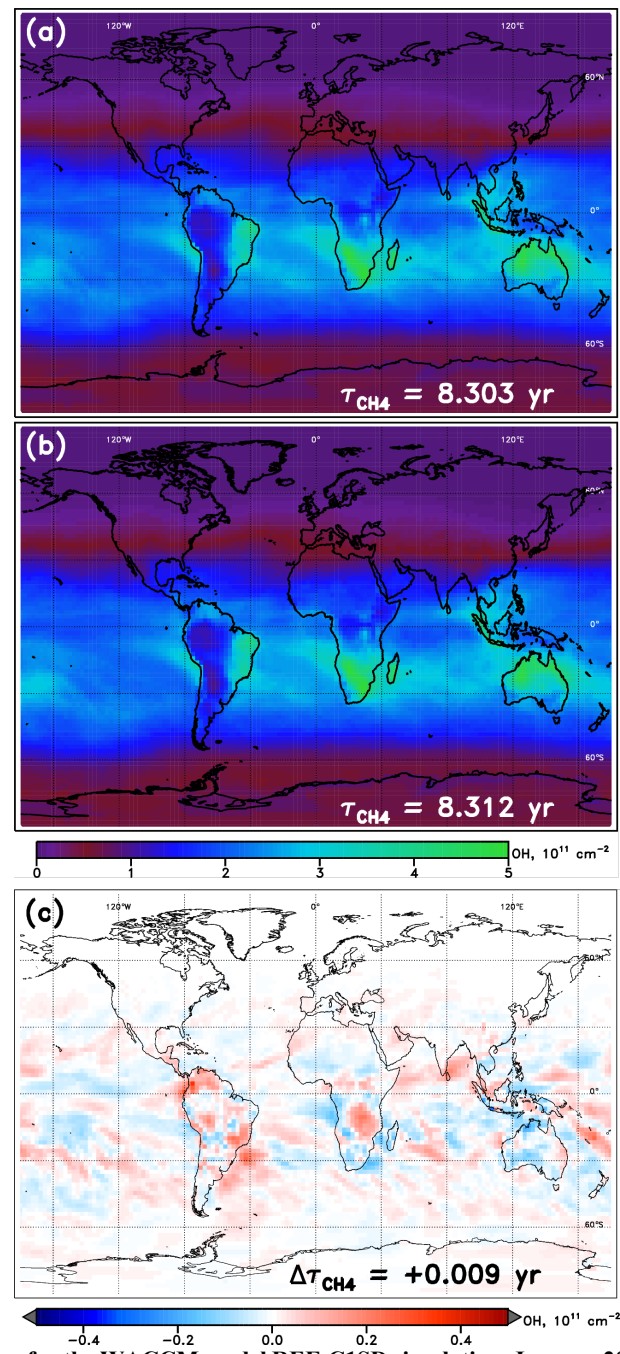

**Figure 3. Tropospheric OH columns for the WACCM model REF-C1SD simulation, January 2000. (a) Columns calculated directly from the WACCM output; (b) columns calculated from the output from the WACCM January NN run with inputs from the native model; (c) difference in column values, (NN – model). Methane lifetime values calculated from 3-D OH fields from WACCM and from the WACCM NN are inscribed in panels (a) and (b), respectively. The methane lifetime difference, (NN – model), is noted in panel (c).**

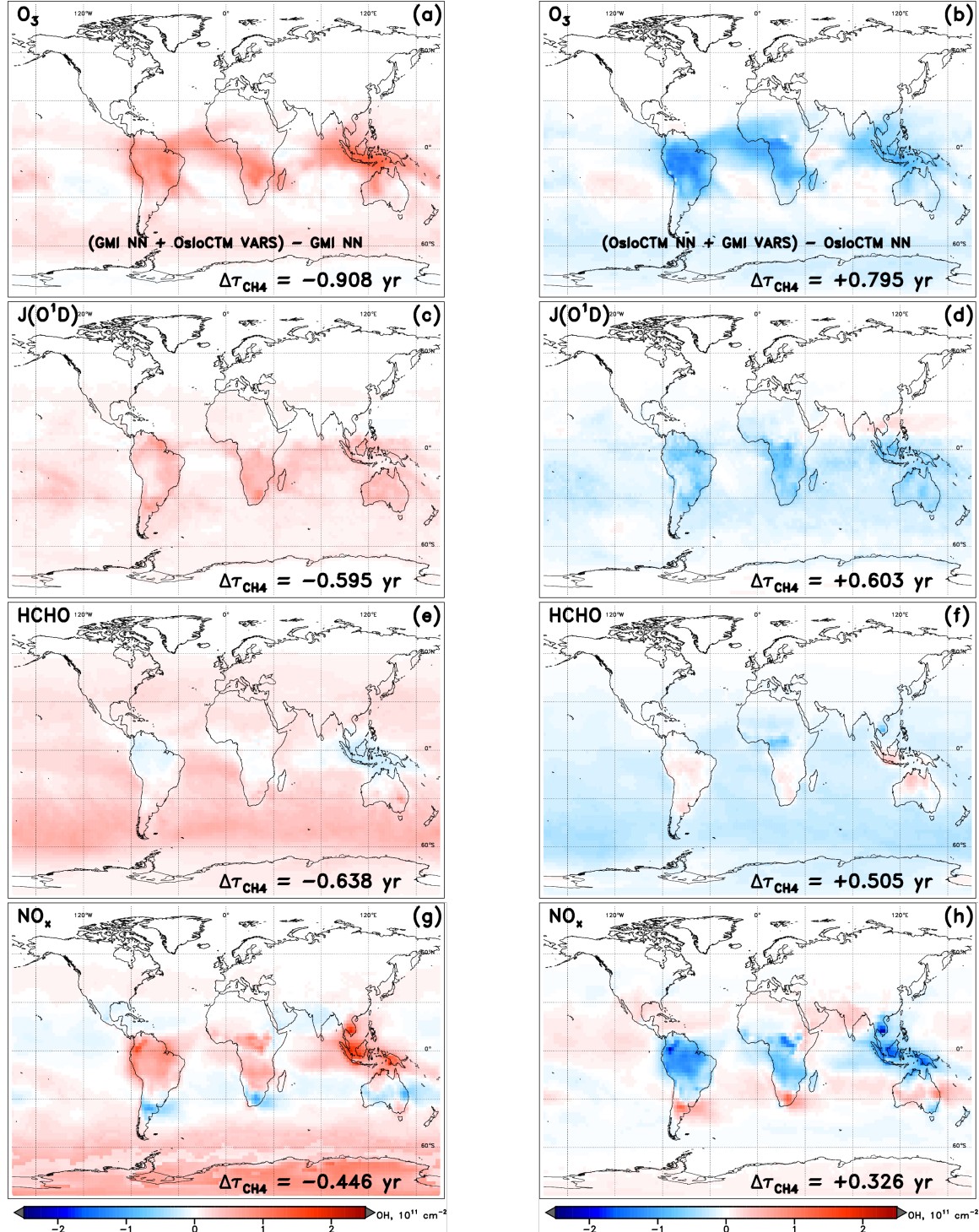

**Figure 4. Changes in tropospheric OH column resulting from swap of indicated variable from another model into the NN of the native model for the specified dynamics simulation of January, 2000. Swaps of the inputs O₃ (a, b), J(O₃→O¹D) (c, d), HCHO (e, f), and NOₓ (g, h) are shown for the GMI (left) and OsloCTM (right) NNs.**

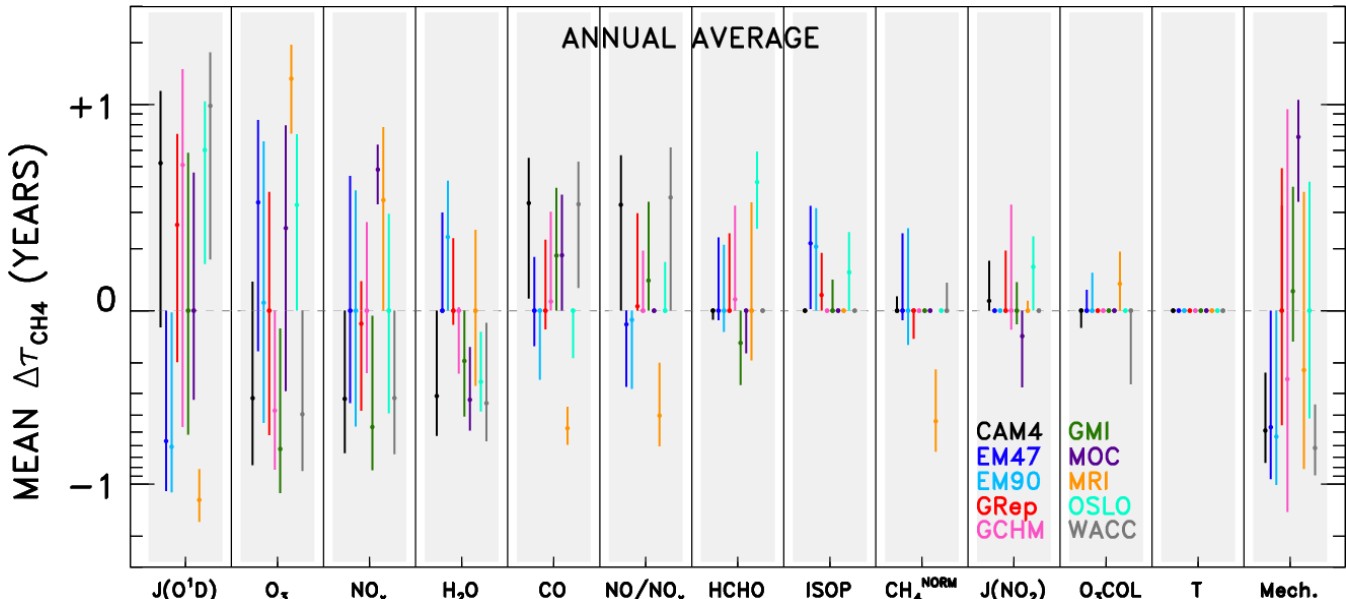

**Figure 5.** Averaged changes in $CH_4$ lifetime accrued for a specified model (color), across all swaps of the indicated variable (x-axis) from all other models. Results are shown annually averaged for year 2000 of the specified dynamics REF-C1SD CCMI and chemical transport model simulations. Circle indicates the mean change in $CH_4$ lifetime; bars represent the 1σ standard deviation from all model pairings. Variables along the x-axis are ranked by averaged magnitude of the $\Delta\tau_{CH4}$ values (i.e., inputs located farther left are responsible for larger differences in $CH_4$ lifetime), except for the "Mech.+Nonlin." term, which is shown last to indicate its role as a remainder term. Model name abbreviations are "CAM4" for CAM4-Chem, "EM47" for EMAC-L47MA, "EM90" for EMAC-L90MA, "GRep" for GEOS Replay, "GCHM" for GEOS-Chem, "GMI" for GMI, "MOC" for MOCAGE, "MRI" for MRI-ESM1r1, "OSLO" for OsloCTM, and "WACC" for WACCM.

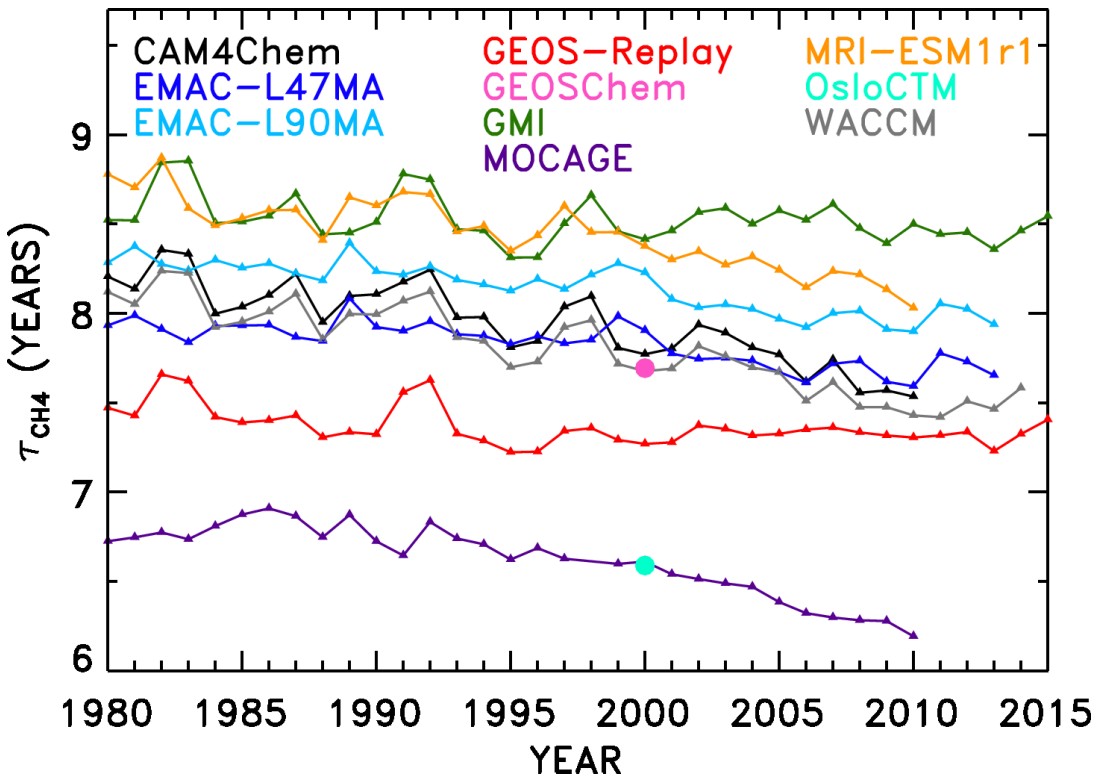

**Figure 6. Time series of CH₄ lifetime from REF-C1SD models. Only one year of output was available for two models (OsloCTM and GEOS-Chem), so their results are shown only as a single data point at year 2000.**

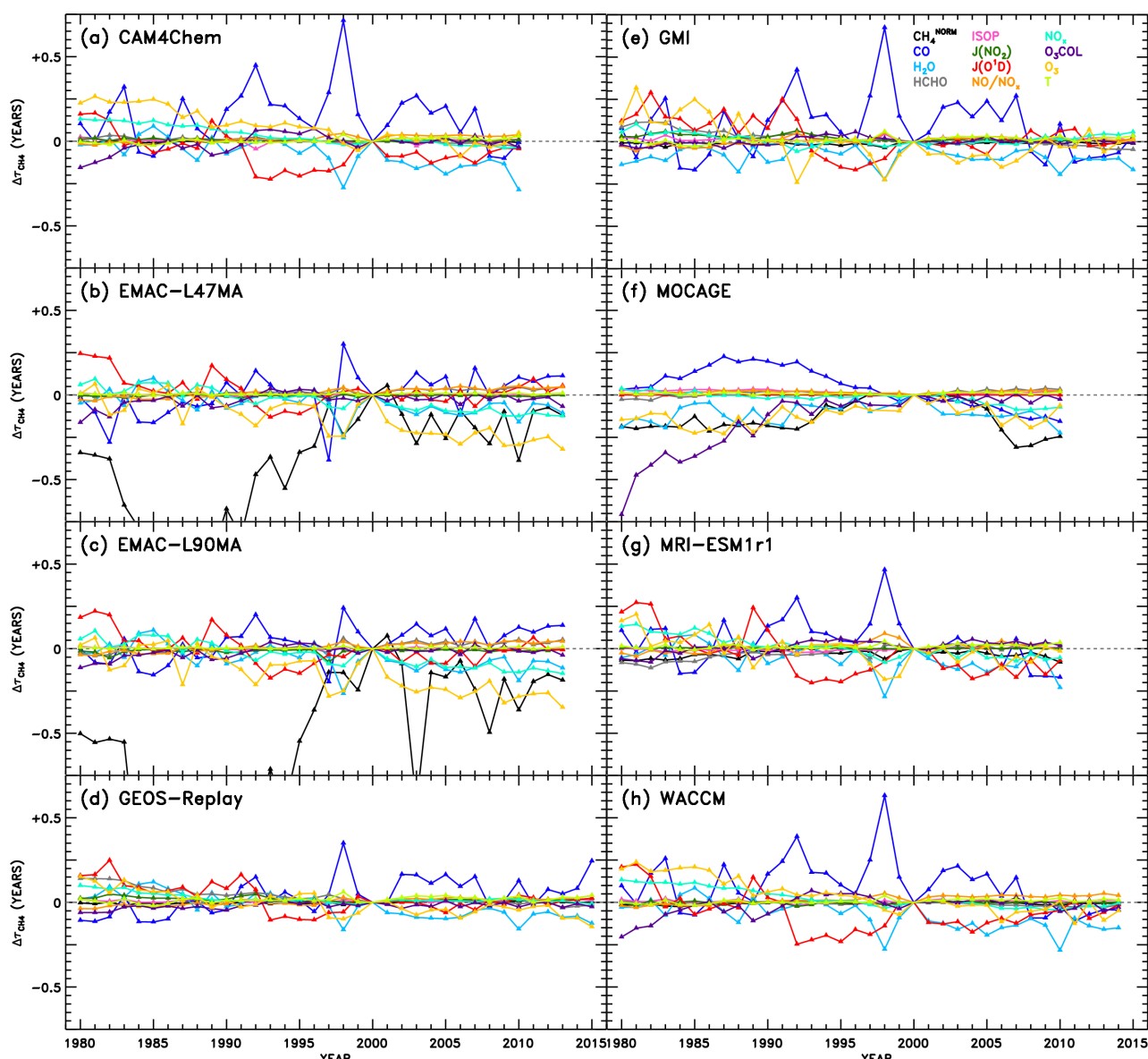

**Figure 7.** Attributions of changes in CH$_4$ lifetime relative to year 2000 of the REF-C1SD simulations. Within the NN of a given model, use of individual inputs (indicated by color) from years other than 2000 result in a change and OH and subsequent CH$_4$ lifetime, shown here. The variations attributable to CH$_4$ are labeled "CH$_4$$^{NORM}$ to designate the use of normalized CH$_4$ fields as inputs to the NNs, as described in Sections 3.1 and 4.3. As a result, OH changes due to CH$_4$$^{NORM}$ represent impacts of changes in how CH$_4$ is distributed within the troposphere, rather than how CH$_4$ concentrations are changing over time.

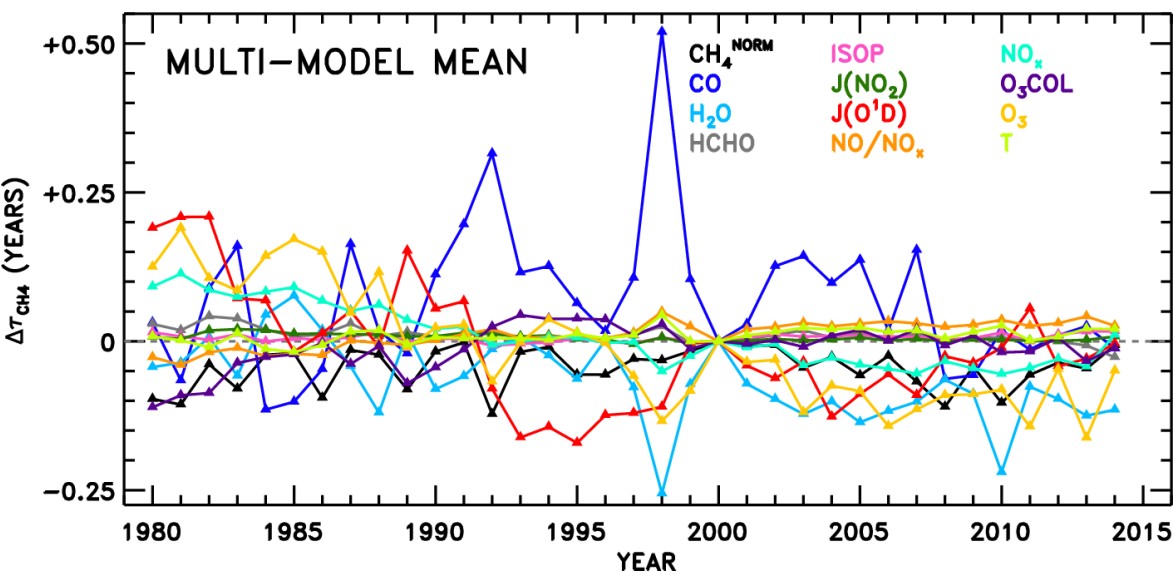

**Figure 8. Same as Figure 7, but the average across all eight models, except filtered to remove NN results for individual months and years during which NN performance is poor, as detailed in the text.**

945

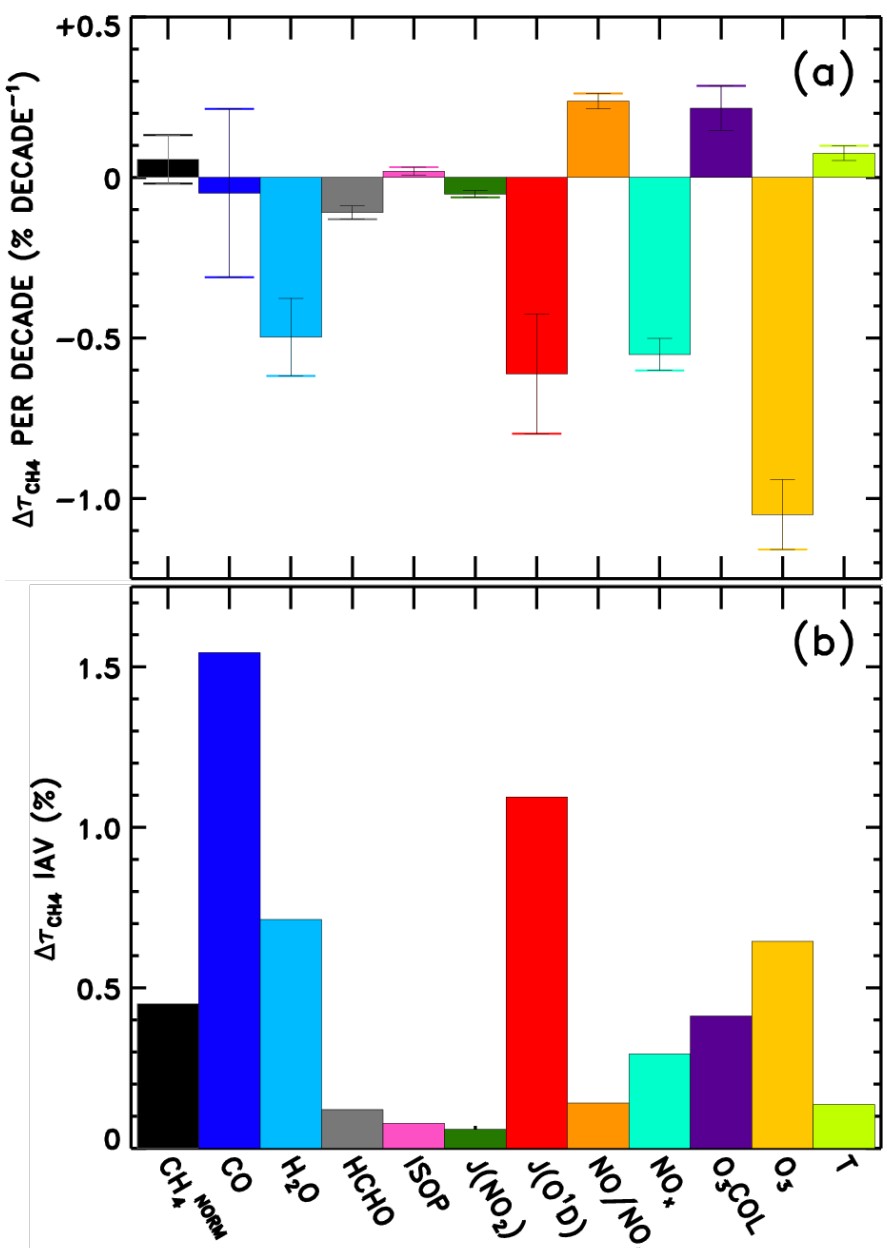

**Figure 9. Multi-model mean linear trend (a) and interannual variability (b) in $\tau_{CH4}$ attributed to each variable examined through the NN method.**

950

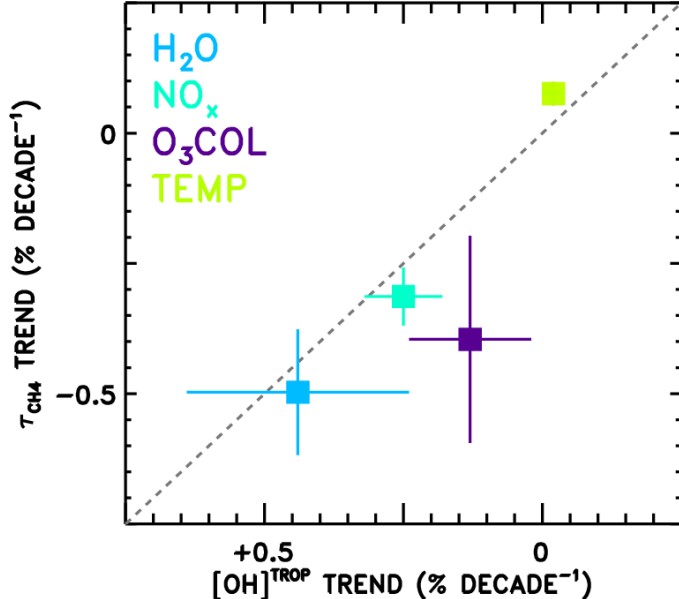

**Figure 10.** Comparison of the attributed trends in $\tau_{CH4}$ found in this work according to the REF-C1SD simulations performed for CCMI (y-axis) to the attributed trends in tropospheric mean OH ("$[OH]^{TROP}$") found based on observations in Nicely et al. [2018]. The grey dashed line indicates the –1:1 line, as values should be anti-correlated. The $\tau_{CH4}$ trend numbers from this work for $NO_x$ combine the $NO_x$ total abundance and partitioning ($NO/NO_x$) values from Figure 9, and for $O_3$ Column combine the $J(O^1D)$ and $O_3$ Column values, as both effects are encompassed in the determination of $[OH]^{TROP}$.