# Peer review of "A Machine Learning Examination of Hydroxyl Radical Differences Among Model Simulations for CCMI-1"

_Atmospheric Chemistry and Physics, 2019_

## Referee Comment (RC1) · Peer Johannes Nowack (Referee) · 2 Oct 2019

The paper by Nicely et al. uses a neural network approach to infer drivers of differences in OH/methane liftetimes among chemistry-climate models. In addition, the approach is used to understand modelled historic trends and variability in these variables. The method itself has been applied in similar form before (cf. Nicely 2017), but here it is applied to a novel set of specified dynamics CCMI simulations.

Overall this paper is a nice example of how machine learning can be used to provide novel insights into chemistry-climate model differences and I enjoyed reading it. I would therefore definitely recommend rapid publication subject some revisions and

clarifications concerning my comments listed below.

Major comments:

- The use of neural nets and especially their cross-validation requires further motivation and explanation. I know this can feel like unnecessary repetition to the authors given that the method has been described previously, but it is an essential aspect due to the central role of the method here. For example, when I first read the paper I was entirely unclear if all results might be subject to overfitting and if the sampling was done in space or time as well as how the data was split into training, cross-validation and test datasets; an essential aspect of any machine learning application. I now understand from reading the other paper that probably regressions were fit on an 80%/10%/10% split of the year 2000, using each grid cell as one sample for a month (rather than samples being ordered by time). Is this still valid? Is early-stopping really the only method you used to manage the bias-variance trade-off? This point is particularly important as evaluation results are given only for the year 2000, which as mentioned is used for training. Given that the year was used for training it would not be surprising if the neural net can fit the data almost arbitrarily well if overfitting wasn't sufficiently counteracted. Maybe show results/evaluate for all years that you did not use for training? I would also explicitly mention the sample size for each dataset (all models are interpolated to the same resolution?).

- I would like an additional explanation of why neural nets were used in the first place. I know they can model complex non-linear functions (which is one point that could be mentioned), but there are many algorithms that can do the same but would probably be more suited for inference tasks such as the one attempted here. Random forests, for example, would immediately provide feature importances for the regression models themselves and it would be easier to test dependencies between correlated variables (e.g. ozone, T, humidity) where it is

unclear what is cause and effect. I do not ask for a refit with different algorithms, but it could be mentioned in terms of future work/context.

- some more reflection on the role of the nudged dynamics: the authors mention that one of the reasons why temperature is less important in explaining inter-model differences is the fixation to a common atmospheric background state by nudging. Alternatively, correlations with other variables such as ozone are offered as an explanation. Could the same not be said about water vapour? Maybe this would also explain why it is suddenly so much more important (relatively) to explain variability? What did you observe in this respect for the free-running simulations?

- the randomness of neural networks: it seems that only one network is fit per model. Unfortunately, neural networks behave somewhat randomly, which is essentially the result of many different local minima in the cost function that can be found during the weight optimization process. Therefore, I would expect that the networks for each model would already be different due to different random initializations of the networks even if the chemistry models would be identical. I would strongly encourage the authors to test the relative importance of this randomness aspect compared to the actual inter-model differences. For example, they could train five-ten neural networks for two of the models (subject to an objective optimization/early stopping procedure) and show the spread in the results when these different network realizations of the two models are compared (instead of only one realization for each). No need to get started with different network architectures, which would similarly affect the results, I assume.

Minor comments:

- p. 4, l.107-109: revise second part of the sentence.

- p. 4/5; model simulations: since UV fluxes and stratospheric ozone are discussed maybe briefly mention if all/which models include interactive stratospheric chemistry, or how it is treated otherwise.

- section 3.1 I think there should be more detail here; essentially another small subsection on the cross-validation method.

- p. 5 l. 161: 'mutually exclusive' - what do you mean by that here?

- l. 165-170: Maybe try a variation of the input features? The cross-correlations are indeed an obvious problem for the interpretation. Did you consider fitting two different networks, e.g. one with JO1D, one with column ozone and consider how well they do on the cross-validation dataset? I am also wondering how these different networks would perform in different atmospheric regimes, e.g. column ozone being more important in the upper troposphere. JO1D (including clouds) becoming relatively more important in the lower troposphere? Can a single network for all grid cells capture these different regimes appropriately?

- l.228: performance for the year 2000 is strong – but this is the training year. Should the goal not be to evaluate on out-of-sample years. Maybe show an error plot for all years? I assume it gets worse the further one moves away from the training year, partly due to the extrapolation error?

- general remark on the extrapolation issues: could you give an estimate of how often you had to correct values in this way for each comparison/model (e.g. percentage o cases depending on the year)? This would give the reader a better impression of how important this factor is when considering the results. In addition, did you ever test how linear/non-linear the regression relationships really are? Maybe linear regression algorithms such as Lasso/Ridge would actually circumvent all these issues by being able to extrapolate better and still extract

feature importances in a sophisticated enough manner (the resulting regressions would also be easier to interpret).

- l. 484: maybe I approach this one too naively, but why would I expect to model a CH4 trend if CH4 is normalized by its maximum value in each year? I assume the maximum value shows a trend somewhat proportional to the average trend?

---

## Referee Comment (RC2) · Anonymous Referee #2 · 3 Oct 2019

Nicely et al. (2019) attributed OH differences among CCMI models into a number of parameters using a neural network approach. They found the major drivers for the decline in methane lifetime are tropospheric O3, JO1D, NOx, and H2O, with CO contributing to the OH interannual variability. It is a very interesting study with very popular machine learning technique. The manuscript is in general well written and well organized. I recommend acceptance of the manuscript after addressing below questions.

Neural network setup

As described in the manuscript, one NN is trained for each model for each month and

all the training is performed for year 2000. So how is it applicable for the input with a lengthy period? Some variables would undergo significant changes from the 1980s to 2010s. What if the NN trained for year 2000 is not suitable for 1980s or 2010s?

There is one concern that when you substitute a single input taken from one model into another. Would this affect the original chemical regime or atmospheric condition? Would there be some "relaxation" in the system to approach original condition? In that sense, it could reduce the sensitivity of OH to the differences in the input.

Lastly, it is more of a broad question. To what degree that the trained NN can realistically represent the non-linear chemical system. In this work, there are a number of variables are input to the NN. The weighting factors can be adjusted during the training process, but if there are more inputs or different inputs, the weighting factors could be different? Would this affect conclusion? How to deal with this issue?

Specific comments:

Page 4, 121-125, is water vapor nudged for all the REF-CISD simulations? If not, what are the REF-C1SD simulations that nudge water vapor?

Page 8, line 244, but also over tropical ocean?

Page 9, line 261-264, could you elaborate "buffering effects"?

Page 9, line 278-282, this is similar to "relaxation" that mentioned in the general comments.

Page 11, line 326 &ff, Figure 5, the impacts of temperatures are small due to the specified dynamics in the model. What about water vapor? If specified water vapor is also imposed, are the impacts of water vapor still large? You may want to check the models with the specified water vapor.

Page 11, line 336-378, what do you mean by "reminder term"?

Page 16, line 487-489, are you talking about latitudinal gradient or vertical distribution?

---

## Short Comment (SC1) · 7 Oct 2019

**Comments: A Machine Learning Examination of Hydroxyl Radical Differences Among Model Simulations for CCMI-1 by Nicely et al., 2019**

**Karl M. Seltzer, Prasad Kasibhatla**

Nicholas School of the Environment, Duke University, Durham, NC, USA

Email: karl.seltzer@duke.edu, psk9@duke.edu

**General Comments**

The manuscript "A Machine Learning Examination of Hydroxyl Radical Differences Among Model Simulations for CCMI-1" by Nicely et al. discusses a topic that is of high interest to the Atmospheric Chemistry and Physics community. Possibly the most perplexing issue in atmospheric chemistry is the unexpected stabilization of global methane concentrations from ~2000-2006. This study attempts to unravel the individual CTM drivers of the hydroxyl radical in a suite of simulations, thus illuminating the changes, and reason for said changes, in the primary termination pathway for methane, as simulated by each CTM.

While this work is important, we do have concerns about how some of the results are presented and methods are employed in this analysis, both of which constitute major comments. We will describe both in more detail below, followed by some minor comments.

**Major Comments**

1. In Figures 7-10, results from the $CH_4$ signal, as it relates to changes in tropospheric OH, are presented. While the text does explicitly state that "$CH_4$" is a normalized value based on the maximum tropospheric value, we believe the presentation of the results in Figures 7-10 and much of the language used throughout the manuscript can lead to substantial confusion on the part of the reader. The reader might reasonably interpret the results as an estimate of the sensitivity of $\tau_{CH4xOH}$ to changes in $CH_4$ abundance (i.e. the $CH_4$ feedback factor). One example: the inclusion of $CH_4$ in Figure 10 makes a comparison of the "$CH_4$" value reported in this study (i.e. NOT the $CH_4$ feedback factor) with the calculated $CH_4$ feedback factor from Nicely et al., 2018.

    Based on our interpretation of the methods employed here, the authors did not analyze the $CH_4$ feedback factor. Since it seems the better characterization is that the global *distributions* of $CH_4$ concentrations were analyzed, we think the authors need to re-write any discussions related to $CH_4$ results throughout the manuscript to make this distinction abundantly more clear, and should possibly remove the characterization of "$CH_4$" in Figures 7-10. Similarly, it is not clear why $CH_4$ concentrations were normalized. Presumably, the same analysis using non-normalized values of $CH_4$ would be able to capture the $CH_4$ feedback?

2. The sensitivity of $\tau_{CH4xOH}$ to changes in $CH_4$ abundance reported by CTM studies are reasonably consistent and range from -0.25 to -0.35 (Prather et al., 2001; Fiore et al., 2009; Holmes et al., 2013, Holmes 2018). That is, the tropospheric OH abundance declines by 0.25%-0.35% for every 1% increase in $CH_4$ abundance (Prather et al., 2001). The IPCC AR5 reported that global $CH_4$ abundance grew by ~13% from 1980 to 2010 (Ciais et al., 2013).

Assuming the models used here respond in a similar manner to other published CTM studies, the $CH_4$ feedback should have yielded a ~3.3%-4.6% decrease in tropospheric OH between 1980-2010 (or equivalently, 1.1%-1.5% per decade). That driver should theoretically be captured in the net results presented in Figure 6.

As noted on Line 457, the mean downward trend in $\tau_{CH4}$ of Figure 6 is 1.8% per decade. Therefore, the residual (i.e. all of the other factors outside of the $CH_4$ feedback) should be ~(-1.8% - 1.3%) → -3.2% per decade (note: 1.3% is the average of 1.1% and 1.5%). This is much larger than the ~residual of -1.9% reported on Line 457 (~residual because it does not include the $CH_4$ feedback factor). Therefore, since the $\tau_{CH4}$ budget does not appear to be closed when adding up all of the variables (including the $CH_4$ feedback), this suggests that the methods used here have difficulty in deriving the contributions of individual drivers. If so, that would be a fundamental issue with the methods used to derive Figures 7-10. Here are some ways we believe the authors can build confidence in the methods used here:

    a. A quick first step would be to add up all of the components for each model in Figure 7 and plot their change, side-by-side, to the values presented in Figure 6 (normalized to 2000 values for consistency). Do the trends match? If yes, since the NN method does not account for the $CH_4$ feedback and CTMs are known to have a robust and consistent $CH_4$ feedback, why do they nonetheless match? If no, can the missing $CH_4$ feedback explain the difference?

    b. A lengthier, but maybe necessary test: Experiment with one of the CTMs. For example, re-run GMI with the year 2000 repeating for all variables, except CO. This might only be necessary for a few select years, such as 1985 and 1998. Do these results match the dark blue line in Figure 7e? One or two examples of these types of validation steps would really increase our confidence in the driver analysis.

    c. When attributing specific, individual drivers to trends, Random Forests are considered better machine learning tools (Grange et al., 2018). It is likely easy to swap out the NN code in your analysis with a random forest. Experiment with one of the models. For example, run the random forest algorithm for GMI's 2000 results and repeat the process for Figure 7. How different are the results?

**Minor Comments**

- Figure 3 compares the tropospheric OH columns from WACCM and the ANN-WACCM predicted tropospheric OH columns. As noted on Line 174, the training methods in this analysis were the same as those carried out in Nicely et al., 2017, which stated that the training/validation/testing datasets comprised 80/10/10% of all data. Therefore, it seems that 80% of the data that was used to construct the middle panel of Figure 3 was data that the ANN has seen before (i.e. from the NN training). Shouldn't this part of the evaluation be restricted to only the testing dataset?
- In the paragraphs spanning Lines 423-448, there is a discussion about "spurious results". Are these results "spurious" just because they look out of place in Fig. 7, or are there some other quantifiable ways that might justify the label "spurious"?

- Figure 9b: Don't CTMs have difficulty in capturing observation-derived estimates of IAV (Holmes et al., 2013)? That should be noted.
- Lines 482-498 should likely be removed. The comparison of the $CH_4$ results here and the $CH_4$ results in Nicely et al., 2018 are not an 'apples-to-apples' comparison, as noted by the authors in the sentence starting with "On one hand…" from Line 485.

**References**

Ciais, P., C. Sabine, G. Bala, et al., 2013: Carbon and Other Biogeochemical Cycles. In: Climate Change 2013: The Physical Science Basis. Contribution of Working Group I to the Fifth Assessment Report of the Intergovernmental Panel on Climate Change. 2013.

Fiore, A. M., Dentener, F. J., Wild, O., et al.,: Multimodel estimates of intercontinental source-receptor relationships for ozone pollution, J. Geophys. Res., 114(D04301), doi:10.1029/2008JD010816, 2009.

Grange, S. K., Carslaw, D. C., Lewis, A. C., et al.,: Random forest meteorological normalisation models for Swiss PM10 trend analysis, Atmos. Chem. Phys., 18(9), 6223–6239, doi:10.5194/acp-18-6223-2018, 2018.

Holmes, C. D.: Methane Feedback on Atmospheric Chemistry: Methods, Models, and Mechanisms, J. Adv. Model. Earth Syst., 10(4), 1087–1099, doi:10.1002/2017MS001196, 2018.

Holmes, C. D., Prather, M. J., Søvde, O. A. and Myhre, G.: Future methane, hydroxyl, and their uncertainties: Key climate and emission parameters for future predictions, Atmos. Chem. Phys., 13(1), 285–302, doi:10.5194/acp-13-285-2013, 2013.

Nicely, J. M., Salawitch, R. J., Canty, T., et al.,: Quantifying the causes of differences in tropospheric OH within global models, J. Geophys. Res., 122(3), 1983–2007, doi:10.1002/2016JD026239, 2017.

Nicely, J. M., Canty, T. P., Manyin, M., et al.,: Changes in Global Tropospheric OH Expected as a Result of Climate Change Over the Last Several Decades, J. Geophys. Res. Atmos., 123(18), 10,774-10,795, doi:10.1029/2018JD028388, 2018.

Prather, M., Ehhalt, D., Dentener, F., et al.,: Atmospheric Chemistry and Greenhouse Gases, in Climate Change 2001: The Scientific Basis. Third Assessment Report of the Intergovernmental Panel on Climate Change, 2001.

---

## Referee Comment (RC3) · Leif Denby (Referee) · 10 Nov 2019

I am only commenting on the machine learning aspect of the submitted manuscript. Apologies for overlooking for not providing more general feedback.

1. In section 3.2 I would rephrase the sentence containing "mimic the tropospheric chemistry" to include "predict the instantaneous OH" concentration. As is written now it might give the impression that the time evolution is predicted by the neural networks as the research presented is about reactions.

2. I find the sentences "Briefly, one NN is trained for one model, for one simulation

month at a time." and "To reduce computational demands, we establish NNs for four months, one for each season..." a little contradictory. Is training done on one month or on four months of input? How is it possible to do both? It might be that the reader should simply study the referenced paper, but I find this a little unclear.

3. It would be nice to a brief comment on why models were trained for each month separately. Was this done because the temporal variability couldn't be captured by a single model? Does the skill of each model vary through the month? I assume that at the ends of the month (where there is transition between which model is used) there might be a reduction in skill. But maybe the predictions match seamlessly when switching between models.

4. The "Inter-model comparison" is nice. With the restriction on the numerical range of the values which are substituted I feel that feature importance could similarly be inferred by simply shuffling (across time) all values for a specific feature, similarly to how it's done for random forests. Is there a reason why this wasn't attempted here? Isn't there a concern that using the presented method that one might infer low feature importance for fields that simply vary little between models?

---

## Author Response (AR1)

**Response to reviewers**

**A Machine Learning Examination of Hydroxyl Radical Differences Among Model Simulations for CCMI-1**

**Nicely et al.**

On behalf of all coauthors, I thank the reviewers and commenters for their time taken to read the manuscript and offer constructive comments. They have served to significantly strengthen the analysis. Below, we address each comment and, where applicable, detail how the manuscript was revised in response. Original reviewer comments are shown in black font, and our responses are shown in blue.

**Reviewer 1 – Dr. Peer Nowack**

The paper by Nicely et al. uses a neural network approach to infer drivers of differences in OH/methane liftetimes among chemistry-climate models. In addition, the approach is used to understand modelled historic trends and variability in these variables. The method itself has been applied in similar form before (cf. Nicely 2017), but here it is applied to a novel set of specified dynamics CCMI simulations.

Overall this paper is a nice example of how machine learning can be used to provide novel insights into chemistry-climate model differences and I enjoyed reading it. I would therefore definitely recommend rapid publication subject some revisions and clarifications concerning my comments listed below. Major comments:

• The use of neural nets and especially their cross-validation requires further motivation and explanation. I know this can feel like unnecessary repetition to the authors given that the method has been described previously, but it is an essential aspect due to the central role of the method here. For example, when I first read the paper I was entirely unclear if all results might be subject to overfitting and if the sampling was done in space or time as well as how the data was split into training, cross-validation and test datasets; an essential aspect of any machine learning application. I now understand from reading the other paper that probably regressions were fit on an 80%/10%/10% split of the year 2000, using each grid cell as one sample for a month (rather than samples being ordered by time). Is this still valid? Is earlystopping really the only method you used to manage the bias-variance trade-off? This point is particularly important as evaluation results are given only for the year 2000, which as mentioned is used for training. Given that the year was used for training it would not be surprising if the neural net can fit the data almost arbitrarily well if overfitting wasn't sufficiently counteracted. Maybe show results/evaluate for all years that you did not use for training? I would also explicitly mention the sample size for each dataset (all models are interpolated to the same resolution?).

We agree with Dr. Nowack that the details of the method are important and have included more description in response. Specifically, we now write in the main text, at **L169**:

"Each model gridbox located below the tropopause (thermal, following the WMO definition, for all models except GEOS Replay, which uses a "blended" tropopause calculation combining thermal and

potential vorticity definitions) is a single sample, so sample sizes are determined by a model's vertical and spatial native resolution. The number of tropospheric model grid points, and thus the training dataset sample size, is indicated for each model in Table S1 and always exceeds 100,000. Because separate NNs are trained for each month, and monthly mean output from each model simulation is used as input and training data, the dataset does not represent diurnal variations in OH chemistry."

**and at L205:**

"For training, the model output is randomly split 80%/10%/10% into training, validation, and test datasets. During that process, the data from the training set is used to actively adjust weighting factors, and the validation set is evaluated to determine a training stopping point. When errors in predicting the validation data grow after adjusting weighting factors some number of iterations in a row, it is determined that the NN model prior to the growth in errors likely reached a local minimum in its cost function. This manner of "early stopping" helps to prevent over-fitting, though application of the NNs to alternative years is not immune to over-fitting, an issue discussed further in Section 4.3.1. For further application of this method across varying time scales, we would recommend a more methodical approach to sampling model output in time as well as in space. The final 10% of data is then used to independently test the resulting NN, and compare between different training iterations. A total of five trainings were performed for each NN, and the NN with best performance (evaluated by the correlation coefficient from comparison of NN-calculated and model-simulated OH values) was chosen as the NN to be used in further analysis."

To further detail the training datasets used here, we have added Table S1 to the Supplement, listing the sample size of the datasets (i.e., number of tropospheric model grid points) used for each model, for each month.

We now include performance metrics of all NNs relative to year 2000, as were used in choosing between training versions. Figures S1-S4 have been added, showing correlations of NN-calculated and model-simulated OH, and new Tables S2 and S3 provide statistics of each NN used in this analysis. Associated text is included at L300.

We have also added evaluation of the NNs' performance for years other than 2000, and have modified our analysis of the time series of CH4 lifetime based on this more quantitative identification of ill-performing (i.e., overfit) NNs (whereas before, we had subjectively removed a few cases of "spurious results" that stood out by eye). We use the evaluation of NN performance for each year as a guide; if the  $r^2$  value of the NN-calculated OH (compared to the native model's OH) is greater than some threshold for a given year, then we will use that NN for that year. If not, that NN will be excluded, for that year. We somewhat arbitrarily decided on an  $r^2$  threshold of 0.95, though we found the resulting variations and trends in CH4 lifetime to be relatively insensitive to varying this threshold within reason. To demonstrate this, we have added the original Figs. 8 and 9, generated without the new quality check, to Supplement as Figs. S24 and S25.

The implementation of this NN quality check changed our results slightly (the trend in  $\tau_{CH_4}$  due to tropospheric O3, for instance, is a bit larger in magnitude), but the major conclusions of the analysis remain unchanged.

We thank the reviewers for suggesting more attention to NN overfitting, and have revised the manuscript at **L605** to describe our modified approach:

"These examples of spurious results highlight an issue that must be treated with caution when using machine learning approaches. Because the application of our NN method to time series analysis is an extension beyond the originally intended purpose, not all NNs are sufficiently generalizable to reliably reproduce OH for years other than the training year, 2000. To account for this, we evaluate each NN for all years by inputting variables from each year. With this test, all inputs are changed, not just a single input at a time. The resulting OH, as depicted in Figures S16-S23 for select years, compares well to the native model's OH field for that year in many cases, but not in all. Considerable bias occurs at low OH mixing ratios, though we note that near-zero concentrations will likely not affect the resulting globally-integrated  $\tau_{CH_4}$  unless values are grossly overestimated. This evaluation also represents a rigorous test of the NNs, as significant shifts in numerous inputs at once might push the NN algorithm into new phase space not encountered during training, much more so than only changing one input at a time, which is our approach in the subsequent time-series analysis. Nonetheless, we limit the influence of poorly generalizable, or "overfit," NNs by only including in the multi-model mean results for the years in which a NN reproduces its native model's OH field with an  $r^2$  value greater than or equal to 0.95. For four NNs (one per month) created for each of 8 CCMI models, across 36 years, the potential application of the NNs to 1152 calculations  $(4 \times 8 \times 36)$  is reduced to 696 calculations using this test. Results from this point forward are subject to this quality check, and were found to be insensitive to the  $r^2$  threshold imposed. This insensitivity is demonstrated by alternate versions of the figures to come, placed in Supplement, generated using all NNs rather than the quality-filtered NNs."

• I would like an additional explanation of why neural nets were used in the first place. I know they can model complex non-linear functions (which is one point that could be mentioned), but there are many algorithms that can do the same but would probably be more suited for inference tasks such as the one attempted here. Random forests, for example, would immediately provide feature importances for the regression models themselves and it would be easier to test dependencies between correlated variables (e.g. ozone, T, humidity) where it is unclear what is cause and effect. I do not ask for a refit with different algorithms, but it could be mentioned in terms of future work/context.

We started this work, as proof-of-concept, ~year 2013 and, since we then had the framework in place to conduct the analysis, we largely adhered to our original method. At that time, we were not aware of the random forest regressions approach, though we have since learned of the technique's benefits, including the feature importance capability. In the event that we are able to continue work in this area in the future, we view it as a high priority to explore the use of alternate techniques, though we remain confident that neural networks are suited to modeling the non-linear aspects of atmospheric chemistry when approached with appropriate caution and quality control measures.

To ensure the reader is aware of these alternative approaches and their similar suitability, we have added text at L223 as follows:

"We note that alternative machine learning algorithms have seen increased application to problems within atmospheric science in the last few years, and may be equally or even better suited than neural networks to studying non-linear chemical systems. In particular random forest regressions and gradient boosting techniques offer greater computational efficiency and, in the case of random forests, have the capability to quickly identify which inputs are most strongly influencing the calculated output, known as "feature importance" (Hu et al., 2017; Keller and Evans, 2019; Liu et al., 2018). ...As such, we encourage exploration of ...additional algorithms for future machine learning applications to atmospheric chemistry."

We have also added text further justifying our use of neural nets, as suggested, at L103:

"NNs in particular are capable of modelling complex non-linear functions, making them a suitable technique for studying the non-linear chemistry involved in OH production and loss."

• some more reflection on the role of the nudged dynamics: the authors mention that one of the reasons why temperature is less important in explaining inter- model differences is the fixation to a common atmospheric background state by nudging. Alternatively, correlations with other variables such as ozone are offered as an explanation. Could the same not be said about water vapour? Maybe this would also explain why it is suddenly so much more important (relatively) to explain variability? What did you observe in this respect for the free-running simulations?

We have added text to the Model Simulations description, stating that most of the specified dynamics simulations do not constrain their water vapor to reanalysis data. While one would think these models would generally calculate water following temperature (which is constrained) as described by the Clausius-Clapeyron relation, this does not seem to be the case, as water appears as a "medium" driver of inter-model differences (4th in the ranking on Figure 5). It's likely that even small differences in water have a large effect on  $\tau_{CH_4}$  due to its important role in primary production of OH.

In the manuscript, we had previously referred to the results of the free-running simulation to interpret this point, at L418:

"We note that T differences between the SD simulations are likely limited due the meteorological constraints imposed on the models. However, examination of the free-running simulations, discussed in the Supplementary Material, also shows practically no impact of T on OH. Thus, we conclude that the effect of temperature on OH chemistry is likely indirect, acting through pathways embodied by other variables, such as H2O and species that exhibit strongly temperature-dependent reaction rates."

We have also added a note regarding Dr. Nowack's point that  $H_2O$  is relatively more important in explaining temporal variability at **L644**:

"It is interesting to note that H2O plays a stronger role in the overall temporal trend of  $\tau_{CH_4}$ , as compared to its role in explaining inter-model differences. This is likely due to the fact that temperatures were constrained in the specified dynamics simulations, which in turn should determine the water vapor calculated within the models."

• the randomness of neural networks: it seems that only one network is fit per model. Unfortunately, neural networks behave somewhat randomly, which is essentially the result of many different local minima in the cost function that can be found during the weight optimization process. Therefore, I would expect that the networks for each model would already be different due to different random initializations of the networks even if the chemistry models would be identical. I would strongly encourage the authors to test the relative importance of this randomness aspect compared to the actual inter-model differences. For example, they could train five-ten neural networks for two of the models (subject to an objective optimization/early stopping procedure) and show the spread in the results when these different network realizations of the two models are compared (instead of only one realization for each). No need to get started with different network architectures, which would similarly affect the results, I assume.

We trained five NNs initially for each model, and choose from among those the top performer, as we now explain in the text (see our response to the first comment, above). We have performed the suggested analysis, reproducing Figure 4 from the main text for five trainings of the GMI and OsloCTM NNs. These reproductions, using different NN versions, have been added to the Supplement as Figures S5 and S6. Visual, qualitative comparison of all 5 versions reveals very similar spatial distributions and magnitudes for the most part, though the values of the change in CH4 lifetime do vary modestly, depending on the variable. For instance, the standard deviations in  $\Delta \tau_{CH_4}$  from JO1D and HCHO swaps into the GMI NNs are about 0.2 years, though some of the NNs included were not as highly-performing as the chosen NN.

In any case, we agree with Dr. Nowack on the importance of the reader being aware of this fact, and have added the following text in our discussion of Figure 4, at L363:

"A fourth issue is the fact that NNs can exhibit some degree of random behaviour, based on how they were trained and initialized. Our method involved training 5 NNs and selecting from those the one that performed best when compared to the independent test dataset. That single NN was used in all subsequent analysis. However, it is a useful exercise to evaluate the role of NN randomness in our results. We show in Figures S5 and S6 the left and right panels of Fig. 4, reproduced for the alternate NN trainings of the GMI and OsloCTM models, respectively. A visual comparison of tropospheric OH column differences among the five trainings of each model's NN reveals markedly similar spatial distributions and magnitudes. The values of calculated  $\Delta \tau_{CH_4}$  do differ somewhat between the training instance, with larger effects on some variable swaps than for others. For instance, the standard deviation of the values of  $\Delta \tau_{CH_4}$  calculated for all five trainings of the GMI NN is about 0.2 years for the J(O1D) and HCHO swaps, but less than 0.05 years for O3 and NOx. We note, though, that some of the NNs displayed in Figures S5 and S6 exhibit worse performance than the one ultimately chosen for subsequent use. As a result of this exercise, the uncertainties resulting from this analysis method may be considered, at most, to be ~0.2 years."

**Minor comments:**

• p. 4, l.107-109: revise second part of the sentence.

We have changed this line from "...seeks to further inter-model evaluation..." to "seeks to enable inter-model evaluation..."

• p. 4/5; model simulations: since UV fluxes and stratospheric ozone are discussed maybe briefly mention if all/which models include interactive stratospheric chemistry, or how it is treated otherwise.

We have included the statement at L137 that "All models, here and including those described below, include interactive stratospheric chemistry."

• section 3.1 I think there should be more detail here; essentially another small subsection on the cross-validation method.

Please refer to our additions to the text described under the first major comment, above. We believe this adds considerably to the detail concerning our method, as Dr. Nowack requests.

• p. 5 l. 161: 'mutually exclusive' - what do you mean by that here?

We have changed the language here to "outside the ranges," to improve clarity.

l. 165-170: Maybe try a variation of the input features? The cross-correlations are indeed an obvious problem for the interpretation. Did you consider fitting two different networks, e.g. one with J01D, one with column ozone and consider how well they do on the cross-validation dataset? I am also wondering how these different networks would perform in different atmospheric regimes, e.g. column ozone being more important in the upper troposphere. J01D (including clouds) becoming relatively more important in the lower troposphere? Can a single network for all grid cells capture these different regimes appropriately?

We did evaluate inclusion of just JO1D (and not column ozone, which was the method of our original "proof-of-concept" study in Nicely et al., 2017), use of just column ozone (and not JO1D), and inclusion of both at the time of original development of the method. While the first and third options were quite comparable in terms of NN performance, the second was not as effective, likely because overhead ozone is so far removed from the *in situ* OH quantity. JO1D, on the other hand, is the immediately-relevant measure of the UV light affecting OH at a particular time and place. Over the years we have worked on this, there has been a substantial amount of experimentation with the input variables chosen, and the set of inputs were determined to provide the best balance between strong performance on independent test data following training, and reasonable results following the inter-model swapping of variables (use of absolute values of CH4 would result in non-sensical output, for instance).

To encourage the reader to consider the importance of architecture/input choice testing, in the case that they attempt a similar analysis, we have added text at **L229** stating:

"We also do not intend to suggest that our chosen NN input list, architecture, and general method is the best approach; input variables were largely determined by available output, and architecture testing was conducted on the computing resources available at the time of the study."

• l.228: performance for the year 2000 is strong – but this is the training year. Should the goal not be to evaluate on out-of-sample years. Maybe show an error plot for all years? I assume it gets worse the further one moves away from the training year, partly due to the extrapolation error?

We refer to our response to the first major comment, where we added evaluation of all NNs for years other than 2000 and adapted our time series analysis to remove poorly-performing NN instances.

• general remark on the extrapolation issues: could you give an estimate of how often you had to correct values in this way for each comparison/model (e.g. percentage o cases depending on the year)? This would give the reader a better impression of how important this factor is when considering the results. In addition, did you ever test how linear/non-linear the regression relationships really are? Maybe linear regression algorithms such as Lasso/Ridge would actually circumvent all these issues by being able to extrapolate better and still extract feature importances in a sophisticated enough manner (the resulting regressions would also be easier to interpret).

The extrapolation control method we developed is only utilized in the inter-model comparison portion of this analysis. Our reasoning for doing so is that any given model should simulate generally comparable conditions from year to year, aside from regime shifts in anthropogenically-emitted species and strong ENSO events that shift locations of convection, biomass burning, etc. And, if our NNs are moderately generalizable, then small excursions outside the range of variable values on which the NNs were trained should be manageable (which we found was not always the case and have now accounted for, as explained under the first comment, above).

We still agree with Dr. Nowack, though, that some indication of how frequently extrapolation control is employed in the inter-model comparison analysis would be informative. To do this, we have written out flags for instances in which adjustments to swapped input variables are made. We separate these instances into "coarse" and "fine" adjustments, the former describing the case when an incoming value falls completely outside the range of tropospheric values from the NN's native model, incurring a presumably large adjustment, and the latter describing smaller changes made to conform the other model's variable to the native model's chemical regimes. For the January NNs only (the extrapolation control code is rather inefficient; it took several weeks to generate these results alone), the percentage of cases (total number of tropospheric grid points) that undergo coarse adjustment are 3.5% on average, while cases in which fine adjustments are performed average 18.8%. A large number of the fine adjustment cases are driven by inconsistencies in CH4, though individual models may have other factors that contribute significantly.

The result of performing these adjustments is to dampen the calculated impact to OH and  $\tau_{CH_4}$ . However, the analysis already reveals instances with large changes to  $\tau_{CH_4}$ , so we think it is appropriate to state more conservative results with a higher level of confidence.

We have added text to inform the reader how often extrapolation control is invoked at L248:

"For reference, we tally the number of instances in which extrapolation control is invoked for two categories: coarse adjustments, when a NN input value from another model falls entirely outside the range of the NN input values from the native model, and fine adjustment, when a value from another model must be tweaked to preserve the native model's chemical regimes. On average, coarse adjustments are incurred for 3.5% of all swapped data points, while fine adjustments are made to

18.8% of swapped values. We find that extrapolation control is critical to achieve meaningful results with the NN inter-model comparison method, though it necessarily forces the attributed changes in OH and  $\tau_{CH_A}$  to be conservative estimates."

We have not explored the use of linear regression techniques such as Lasso/Ridge in order to ameliorate issues pertaining to extrapolation, but we would like to encourage the reader to do so. We have added text at L228:

"Additionally, linear regression algorithms such as Ridge and Lasso regression may be beneficial in curbing issues related to extrapolation."

• l. 484: maybe I approach this one too naively, but why would I expect to model a CH4 trend if CH4 is normalized by its maximum value in each year? I assume the maximum value shows a trend somewhat proportional to the average trend?

Dr. Nowack is correct on this point; we cannot attribute any meaning regarding the "CH4 feedback factor" to the trend in  $\tau_{CH_4}$  due to CH4 found here, as a result of CH4 being normalized. We have taken steps to remove language suggesting that a trend in OH due to CH4 is found, emphasized in some figures that CH4 is normalized by using the notation "CH4NORM", and removed the trend due to CH4 data point from our final figure, comparing the CCMI model trends in  $\tau_{CH_4}$  to our previous empirical study's trends in [OH]TROP, since they are not comparable.

We chose to leave "CH4NORM" in Figs. 7-9 because there is *some* meaning in this value; it represents changes in the spatial and vertical distribution of CH4 within the troposphere, which, based on how the CH4 collocates with high OH concentrations, can influence the resulting  $\tau_{CH_4}$  value. We have added explanation to this effect at L497, where we state:

"Because we are relying on the same NNs used for the inter-model analysis, we emphasize that the CH4 fields used here are still normalized, separately for each year. As a result, the variations in  $\tau_{CH_4}$  due to CH4 should not be interpreted as a measure of the CH4 feedback factor (Prather et al., 2001). Instead of representing the change in OH with a change in absolute concentration of CH4, the numbers shown here signify the change in OH with a change in how CH4 is distributed within the atmosphere. Largely, one would expect this to remain constant over time, though results from this analysis of the CCMI simulations suggests there are some modest changes in  $\tau_{CH_4}$  attributed to the distribution of tropospheric CH4. Should a similar method be applied to analysis of temporal variations in OH in the future, we would encourage training the machine learning algorithm on data spanning all years such that use of CH4 absolute values would be possible."

**Reviewer 2**

Nicely et al. (2019) attributed OH differences among CCMI models into a number of parameters using a neural network approach. They found the major drivers for the decline in methane lifetime are tropospheric O3, JO1D, NOx, and H2O, with CO contributing to the OH interannual variability. It is a very interesting study with very popular machine learning technique. The manuscript is in

general well written and well organized. I recommend acceptance of the manuscript after addressing below questions.

Neural network setup

As described in the manuscript, one NN is trained for each model for each month and all the training is performed for year 2000. So how is it applicable for the input with a lengthy period? Some variables would undergo significant changes from the 1980s to 2010s. What if the NN trained for year 2000 is not suitable for 1980s or 2010s?

This is certainly a concern, and we have added new analysis that identifies how well the NNs, trained on output from year 2000, perform for other years. The majority of NNs continue to perform strongly, though a number of them encounter conditions that cause large deviations in their predicted OH.

We refer the reviewer to our response to Dr. Nowack's first major comment, above, for an indepth description about how the manuscript has been changed to address this point, including adjusted analysis, new methodological details in the main text, and many new figures in the Supplement.

There is one concern that when you substitute a single input taken from one model into another. Would this affect the original chemical regime or atmospheric condition? Would there be some "relaxation" in the system to approach original condition? In that sense, it could reduce the sensitivity of OH to the differences in the input.

We agree with the reviewer, that this analysis neglects "true atmospheric behavior," as you might call it – feedbacks and relaxation effects are ignored as we instead are calculating an instantaneous change that would occur if you could magically perturb a single species or variable. But we would still regard this as a useful exercise to both parse the main influencers of OH chemistry and identify the causes of inter-model differences, which are difficult to do otherwise. We have added text acknowledging this point at L476:

"A final qualification is this analysis constitutes a foundationally hypothetical experiment. It essentially addresses the questions, "What if we could instantaneously switch the fields of just one chemical species between two global models? What would be the impact on OH? on  $\tau_{CH_4}$ ?" This approach, then, necessarily neglects the roles of feedbacks in the atmospheric system (e.g., if the NOx field is perturbed, this will propagate to changes in O3 as well, with time). However, for the objective of teasing apart the influences on global OH abundance and  $\tau_{CH_4}$  and explaining inter-model differences, a notoriously difficult task, we regard our approach as a valuable exercise."

Lastly, it is more of a broad question. To what degree that the trained NN can realistically represent the non-linear chemical system. In this work, there are a number of variables are input to the NN. The weighting factors can be adjusted during the training process, but if there are more inputs or different inputs, the weighting factors could be different? Would this affect conclusion? How to deal with this issue? Neural networks are generally highly capable of modeling non-linear functions, which is the main reason why we chose this approach originally. Within the NN architecture, we use hyperbolic tangent activation functions, which are non-linear. As these functions are used many times over, in parallel, they enable complex fitting of multi-dimensional, non-linear surfaces. The reviewer is correct, though, that once the training process is complete, the weights of our chosen NN are fixed, and the insertion of different inputs can cause issues. To deal with this, we implement the "extrapolation control" method described in the text. In our revisions, we have also taken the extra step of evaluating our NNs across all years, to exclude NNs that do not generalize well (i.e., reproduce well the OH for a particular year, presumably due to some new conditions encountered) from our analysis.

**Specific comments:**

Page 4, 121-125, is water vapor nudged for all the REF-CISD simulations? If not, what are the REF-CISD simulations that nudge water vapor?

We have added statements to the text describing which models include specific humidity nudging in their specified dynamics schemes, at L131:

"Particularly relevant to this analysis is the nudging of specific humidity, which is only performed in the MOCAGE model, of the models we analysed."

**and L156:**

"All CTMs except GEOS-Chem calculate water vapor interactively in the troposphere. GEOS-Chem instead uses specific humidity fields from the MERRA reanalysis."

**Page 8, line 244, but also over tropical ocean?**

Yes, it is true that changes to  $O_3$  and  $NO_x$  influence OH over the tropical oceans as well as over the continents, though the maximum changes in OH appear over land. We have changed this statement to read, at **L319**:

"...exert the greatest influence on OH over the climatological tropics, with maximum impacts over land but extending over the oceans as well."

**Page 9, line 261-264, could you elaborate "buffering effects"?**

The buffering effects we are referring to mostly involve more complex hydrocarbon chemistry, and so we have changed this example from CH4 to isoprene. The text now reads, at **L337**:

"For example, one model may be sensitive to an increase in isoprene, causing OH concentrations to drop in response. Another model may incorporate buffering effects (such as reactions involving oxidized volatile organic compounds (Lelieveld et al., 2016; Taraborrelli et al., 2012) that allow OH to be recycled..."

Page 9, line 278-282, this is similar to "relaxation" that mentioned in the general comments.

The example given at this location, regarding the implementation of extrapolation control preventing a large change in  $CH_4$  from being conveyed to the NN, does not exactly represent a "relaxation" of the system, but rather, a logistical issue with the method, preventing our even testing a large perturbation in the NN out of an abundance of caution. In general, though, we agree with the reviewer that the issue of relaxation is not directly addressed by our method, and so the text added at **L476** (described in our response to the general comment, above) discusses this point.

Page 11, line 326 &ff, Figure 5, the impacts of temperatures are small due to the specified dynamics in the model. What about water vapor? If specified water vapor is also imposed, are the impacts of water vapor still large? You may want to check the models with the specified water vapor.

This is a very interesting idea, though the only two models that used specified water vapor (MOCAGE and GEOS-Chem) used two different reanalysis data sources (ERA-I and MERRA, respectively). The mean  $\Delta \tau_{CH4}$  values due to H2O for these two models are quite different, though there is at least overlap in the 1 $\sigma$  about the mean, represented by the "whiskers" in Fig. 5. Had there been more models imposing a water vapor constraint, from the same reanalysis data, this would be worth exploring further.

**Page 11, line 336-378, what do you mean by "reminder term"?**

We describe the "remainder term" three paragraphs prior to the location identified by the reviewer, where it is instructive to refer to Table 1. To improve clarity, we now include the text: "...the remainder term (or term labelled "Mech." in Table 1)..." at L428.

**Page 16, line 487-489, are you talking about latitudinal gradient or vertical distribution?**

Because of use of normalized CH4 prevents inferences regarding the true "CH4 feedback" on OH, we have removed the discussion at Pg. 16, line 487. However, new text placed earlier in the discussion (starting **L497**) also refers to the "distribution of methane." We have included "both vertically and spatially" as clarification, as changes in the collocation of OH and CH4, no matter where in space, could impact the calculated  $\tau_{CH4}$ .

**Reviewer 3 – Dr. Leif Denby**

I am only commenting on the machine learning aspect of the submitted manuscript. Apologies for overlooking for not providing more general feedback.

1. In section 3.2 I would rephrase the sentence containing "mimic the tropospheric chemistry" to include "predict the instantaneous OH" concentration. As is written now it might give the impression that the time evolution is predicted by the neural networks as the research presented is about reactions.

We agree with Dr. Denby on this use of language and have made the recommended change.

2. I find the sentences "Briefly, one NN is trained for one model, for one simulation month at a time." and "To reduce computational demands, we establish NNs for four months, one for each season..." a little contradictory. Is training done on one month or on four months of input? How is it possible to do both? It might be that the reader should simply study the referenced paper, but I find this a little unclear.

We apologize for the confusing language here; training is done on one month of input; we generate separate NNs for each month that we look at; and we look at 4 months. We have attempted to clarify this by changing the text to read, at **L166**:

"Briefly, four NNs are trained for one model, each for one simulation month. To reduce the computational demands of NN training, we only establish NNs for four months, one for each season..."

3. It would be nice to a brief comment on why models were trained for each month separately. Was this done because the temporal variability couldn't be captured by a single model? Does the skill of each model vary through the month? I assume that at the ends of the month (where there is transition between which model is used) there might be a reduction in skill. But maybe the predictions match seamlessly when switching between models.

In our early years of developing this method, we encountered a couple of issues that resulted in our decision to only train an NN for a single month. When we first attempted to ingest all model output for an entire year into the Matlab NN software, on which we still rely, and on a graduate student's laptop (albeit a powerful one – on which we do *not* still rely), we unsurprisingly encountered memory issues when attempting to train the NN. Then, during our limited attempts to randomly sample model output across all months to generate a training dataset, we found that the NNs did not perform well.

Now that considerably more progress and application of machine learning to scientific questions has taken place, we would encourage a more methodical and strategic sampling of the model domain to create a training dataset. To ensure the reader is aware of this point, we have added text at L231:

"It is possible that a single NN could suffice for predicting OH variations throughout an entire year, rather than for just a single month, following methodical subsampling methods to create the initial training dataset."

Regarding questions about how the NN performs "through the month," we use only monthly mean output from the CCMI models examined here (monthly mean output is commonly what is made available from these large model intercomparison projects). So, we are not able to address issues of varying performance throughout a simulated month.

4. The "Inter-model comparison" is nice. With the restriction on the numerical range of the values which are substituted I feel that feature importance could similarly be inferred by simply shuffling (across time) all values for a specific feature, similarly to how it's done for random forests. Is there a reason why this wasn't attempted here? Isn't there a concern that using the presented method that one might infer low feature importance for fields that simply vary little between models?

We thank Dr. Denby for the statement of support. His follow-up questions relate to the ultimate goal of the analysis. In our case, the objective is to explain why global models of atmospheric chemistry give different quantities of global mean OH and  $\tau_{CH_4}$ . In that case, we are less concerned with quantities that are quite consistent among the models, even if they have the capability of strongly altering OH chemistry. We go into this a bit in the Discussion (~L354), explaining that two conditions must be met to incur a change in OH: differences in the input between the two models, and sensitivity of OH to that input.

The questions posed above would be interesting to address if one were examining which inputs have strong "feature importance," which is not necessarily the goal of the inter-model comparisons, but is, in essence, what we have done in the time series evaluation.

**Short Comment 1 - Mr. Karl M. Seltzer, Dr. Prasad Kasibhatla**

**General Comments**

The manuscript "A Machine Learning Examination of Hydroxyl Radical Differences Among Model Simulations for CCMI-1" by Nicely et al. discusses a topic that is of high interest to the Atmospheric Chemistry and Physics community. Possibly the most perplexing issue in atmospheric chemistry is the unexpected stabilization of global methane concentrations from ~2000-2006. This study attempts to unravel the individual CTM drivers of the hydroxyl radical in a suite of simulations, thus illuminating the changes, and reason for said changes, in the primary termination pathway for methane, as simulated by each CTM.

While this work is important, we do have concerns about how some of the results are presented and methods are employed in this analysis, both of which constitute major comments. We will describe both in more detail below, followed by some minor comments.

**Major Comments**

1. In Figures 7-10, results from the CH4 signal, as it relates to changes in tropospheric OH, are presented. While the text does explicitly state that "CH4" is a normalized value based on the maximum tropospheric value, we believe the presentation of the results in Figures 7-10 and much of the language used throughout the manuscript can lead to substantial confusion on the part of the reader. The reader might reasonably interpret the results as an estimate of the sensitivity of  $\tau_{CH4xOH}$  to changes in CH4 abundance (i.e. the CH4 feedback factor). One example: the inclusion of CH4 in Figure 10 makes a comparison of the "CH4" value reported in this study (i.e. NOT the CH4 feedback factor) with the calculated CH4 feedback factor from Nicely et al., 2018.

Based on our interpretation of the methods employed here, the authors did not analyze the  $CH_4$  feedback factor. Since it seems the better characterization is that the global *distributions* of  $CH_4$  concentrations were analyzed, we think the authors need to re-write any discussions related to  $CH_4$  results throughout the manuscript to make this distinction abundantly more

clear, and should possibly remove the characterization of "CH4" in Figures 7-10. Similarly, it is not clear why CH4 concentrations were normalized. Presumably, the same analysis using non-normalized values of CH4 would be able to capture the CH4 feedback?

We fully acknowledge that the impact of CH4 on the trend in  $\tau_{CH_4}$ , as it is found here, does not represent the CH4 feedback factor. This was a late realization, and some of the language and figures in the manuscript may have been misleading as a result. We have taken steps to remove this misleading content in the following ways:

• During early discussion of the "Time series evaluation" results, we attempt to present this issue in a forthright manner. Starting at L497, the text now reads:

"Because we are relying on the same NNs used for the inter-model analysis, we emphasize that the CH4 fields used here are still normalized, separately for each year. As a result, the variations in  $\tau_{CH_4}$  due to CH4 should not be interpreted as a measure of the CH4 feedback factor (Prather et al., 2001). Instead of representing the change in OH with a change in absolute concentration of CH4, the numbers shown here signify the change in OH with a change in how CH4 is distributed within the atmosphere, both vertically and spatially. Largely, one would expect this to remain constant over time, though results from this analysis of the CCMI simulations suggests there are some modest changes in  $\tau_{CH_4}$  attributed to the distribution of tropospheric CH4. Should a similar method be applied to analysis of temporal variations in OH in the future, we would encourage training the machine learning algorithm on data spanning all years such that use of CH4 absolute values would be possible."

- In Figures 7-9, we now label the time series/trends due to CH4 as "CH4NORM" to serve as a reminder that the CH4 with which we performed the analysis is normalized. We chose to leave "CH4NORM" in Figs. 7-9 because there is *some* meaning in this value; it represents changes in the spatial and vertical distribution of CH4 within the troposphere, which, based on how the CH4 collocates with high OH concentrations, can influence the resulting  $\tau_{CH_4}$  value.
- We have removed entirely the CH4 data point in Figure 10 comparing the CCMI model trend, as evaluated by NN, to the Nicely et al. (2018) trend, and all discussion associated with it, as the two values do not provide an "apples to apples" comparison.

Regarding why normalized CH4 was used in the first place, we sought to utilize the same NNs trained for the inter-model comparison application for the new analysis of OH time series. Absolute values of CH4 mixing ratio as NN inputs were initially attempted for the inter-model comparison, but yielded non-sensical results since the models calculated very different CH4 fields in some cases. In the case of CCMI, the models are fairly similar, since they use the same CH4 boundary condition, but the external models that did not formally participate in CCMI still pose the same problem.

In the case that we could dedicate considerably more time to this work, we would ideally train new NNs for the time series analysis portion of the project using absolute CH4 mixing ratios. This would necessitate that we create a training data set consisting of samples across all years, lest we run into the same dilemma of having the NN trained on a relatively narrow range of CH4 values. This type of subsampling should be performed strategically, and, along with the actual training of the NNs, would be computationally demanding and require a substantial amount of time, thus we regard this as beyond the scope of our current manuscript.

2. The sensitivity of  $\tau_{CH4xOH}$  to changes in CH4 abundance reported by CTM studies are reasonably consistent and range from -0.25 to -0.35 (Prather et al., 2001; Fiore et al., 2009; Holmes et al., 2013, Holmes 2018). That is, the tropospheric OH abundance declines by 0.25%-0.35% for every 1% increase in CH4 abundance (Prather et al., 2001). The IPCC AR5 reported that global CH4 abundance grew by ~13% from 1980 to 2010 (Ciais et al., 2013).

Assuming the models used here respond in a similar manner to other published CTM studies, the CH4 feedback should have yielded a  $\sim 3.3\%$ -4.6% decrease in tropospheric OH between 1980-2010 (or equivalently, 1.1%-1.5% per decade). That driver should theoretically be captured in the net results presented in Figure 6.

As noted on Line 457, the mean downward trend in  $\tau_{CH4}$  of Figure 6 is 1.8% per decade. Therefore, the residual (i.e. all of the other factors outside of the CH4 feedback) should be ~(-1.8% - 1.3%)  $\rightarrow$  -3.2% per decade (note: 1.3% is the average of 1.1% and 1.5%). This is much larger than the ~residual of -1.9% reported on Line 457 (~residual because it does not include the CH4 feedback factor). Therefore, since the  $\tau_{CH4}$  budget does not appear to be closed when adding up all of the variables (including the CH4 feedback), this suggests that the methods used here have difficulty in deriving the contributions of individual drivers. If so, that would be a fundamental issue with the methods used to derive Figures 7-10. Here are some ways we believe the authors can build confidence in the methods used here:

- a. A quick first step would be to add up all of the components for each model in Figure 7 and plot their change, side-by-side, to the values presented in Figure 6 (normalized to 2000 values for consistency). Do the trends match? If yes, since the NN method does not account for the CH4 feedback and CTMs are known to have a robust and consistent CH4 feedback, why do they nonetheless match? If no, can the missing CH4 feedback explain the difference?
- b. A lengthier, but maybe necessary test: Experiment with one of the CTMs. For example, re-run GMI with the year 2000 repeating for all variables, except CO. This might only be necessary for a few select years, such as 1985 and 1998. Do these results match the dark blue line in Figure 7e? One or two examples of these types of validation steps would really increase our confidence in the driver analysis.
- c. When attributing specific, individual drivers to trends, Random Forests are considered better machine learning tools (Grange et al., 2018). It is likely easy to swap out the NN code in your analysis with a random forest. Experiment with one of the models. For example, run the random forest algorithm for GMI's 2000 results and repeat the process for Figure 7. How different are the results?

We acknowledge that these suggestions by Mr. Seltzer and Dr. Kasibhatla would make a rigorous test for the application of our method to time series and determination of trends in  $OH/\tau_{CH4}$ . We have taken steps to build confidence that the method is fundamentally sound following their item (a.) above.

Below we have created a table listing the overall trends in  $\tau_{CH4}$ , taken directly from the CCMI models (i.e., Figure 6), the overall trend calculated by totaling each component from the NN analysis (Figure 7, excluding the obviously spurious cases discussed in the text: EMAC CH4 and MOCAGE O3 Column), and the implied CH4 feedback factor found by subtracting the latter from the former.

| Model      | Native model trend in | Summed trend in $\tau_{CH4}$ | Implied CH 4 |
|------------|-----------------------|------------------------------|-------------------------|
|            | $\tau_{CH4}$ (Fig. 6) | from NN-calculated           | feedback (Column 2      |
|            | (% decade $^{-1}$ )   | components (Fig. 7)          | – Column 3)             |
| CAM4Chem   | -2.69                 | -3.15                        | +0.46                   |
| EMAC-L47MA | -1.22                 | -1.32                        | +0.10                   |
| EMAC-L90MA | -1.48                 | -1.50                        | +0.02                   |
| GEOSCCM    | -0.70                 | -1.57                        | +0.87                   |
| GMI        | -0.54                 | -1.86                        | +1.32                   |
| MOCAGE     | -2.97                 | +1.32                        | -4.29                   |
| MRI-ESM1r1 | -2.31                 | -2.34                        | +0.03                   |
| WACCM      | -2.72                 | -2.97                        | +0.25                   |

The values of the implied CH4 feedback are all of the correct sign, except for MOCAGE, which generally demonstrates quite different behavior from the other models. The value for GMI is in good agreement with the 1.1-1.5 % decade-1 range that is cited in the comment above. We would identify a couple of issues with validating the method in this manner, though.

First, as described in our previous responses to reviewers, we now identify specific instances in which the NNs (for specific months and years) do not perform sufficiently well, and so the multimodel mean trend results that we show now use NN calculations "filtered" for only the high performing NNs. The numbers we quote in the above table, in Column 3, include all NN results except for the MOCAGE O3 column and EMAC CH4 contributions, and so admittedly include some dubious contributions. Because NNs for individual months are filtered out in the new quality-check, it would not be straightforward to calculate new  $\Delta \tau_{CH4}$  values, on a year-by-year basis, for a single model. The aggregation of all models, into the multi-model mean results we present, allows us to assess the trends using results from all years.

Second, we do only perform this analysis for four months out of the year, so a truly apples to apples comparison with, e.g., CH4 feedback factors from other studies would more aptly include all 12 months. And finally, as we now acknowledge following our response to Reviewer 2, there are secondary effects and "relaxation" that occur in the real atmosphere and in the global models we are examining, which the NNs may not capture. This analysis can instead be interpreted as evaluation of the instantaneous change in OH resulting from a hypothetical perturbation to a single chemical/radiative/physical variable. One should be aware that these perturbations often do not occur in isolation, though, and so we now caution the reader that the responses shown by this analysis may not be directly applicable to the real world.

Because we have chosen to remove discussion of our results regarding the trends in  $\tau_{CH4}$  due to CH4, we consider further analysis regarding the CH4 feedback factor as beyond the scope of the current work. We do encourage further study of the issue, though, both by suggesting

refinements to our method (i.e., creating a training dataset sampled across many years, to enable use of absolute CH4 values in the NNs  $\sim$ L223) and by endorsing movement away from CH4 boundary conditions (which we believe hampers useful studies of the CH4 budget with our present-day atmospheric chemistry models), toward interactive fluxes (L687).

**Minor Comments**

• Figure 3 compares the tropospheric OH columns from WACCM and the ANN-WACCM predicted tropospheric OH columns. As noted on Line 174, the training methods in this analysis were the same as those carried out in Nicely et al., 2017, which stated that the training/validation/testing datasets comprised 80/10/10% of all data. Therefore, it seems that 80% of the data that was used to construct the middle panel of Figure 3 was data that the ANN has seen before (i.e. from the NN training). Shouldn't this part of the evaluation be restricted to only the testing dataset?

While the actual evaluation of the NN post-training was performed on the testing dataset, we also wanted in Figure 3 to convey the spatial distribution and magnitudes of tropospheric OH column amounts, relevant for the interpretation of Figure 4. Since our NNs calculate OH on a 3-D basis, and we then integrate the columns in post-processing, generation of a similar figure showing only the 10% of model output used for training would not be possible (i.e., you likely wouldn't have a full vertical profile of OH for any single lat/lon coordinate).

We have, however, added considerable content displaying the performance of our NNs in the form of 2-D histograms in the Supplement, including for years other than 2000 (Figs. S16-S23). While we have newly adapted our analysis to rely only on the NNs that continue to show strong performance in reproducing its native model's OH for a given year, the overall conclusions have changed little as a result (see our response to the first comment by Dr. Nowack for further detail).

• In the paragraphs spanning Lines 423-448, there is a discussion about "spurious results". Are these results "spurious" just because they look out of place in Fig. 7, or are there some other quantifiable ways that might justify the label "spurious"?

We initially identified these "spurious results" by eye, but since looking more closely at the performance of our NNs across all years, we have instituted a quantitative threshold to determine when results from a particular NN/year should be disregarded. We choose a somewhat arbitrary  $r^2$  threshold for this purpose, but we did test the effects of altering this threshold and found little change in our results (again, see our response to Dr. Nowack for further detail).

• Figure 9b: Don't CTMs have difficulty in capturing observation-derived estimates of IAV (Holmes et al., 2013)? That should be noted.

We have added text noting this in our discussion of Figure 9b, at L637:

"The interannual variability of  $\tau_{CH_4}$  is also calculated as the standard deviation of the detrended time series, shown in Fig. 9b, though it is relevant to note that CTMs have historically not captured the full interannual variability exhibited by observed OH proxies (Holmes et al., 2013)."

• Lines 482-498 should likely be removed. The comparison of the CH4 results here and the CH4 results in Nicely et al., 2018 are not an 'apples-to-apples' comparison, as noted by the authors in the sentence starting with "On one hand..." from Line 485.

We concur; this text has been removed.

Correspondence to: Julie M. Nicely (julie.m.nicely@nasa.gov)

Abstract. Hydroxyl radical (OH) plays critical roles within the troposphere, such as determining the lifetime of methane (CH4), yet is challenging to model due to its fast cycling and dependence on a multitude of sources and sinks. As a result, the reasons for variations in OH and the resulting CH4 lifetime ( $\tau_{CH_4}$ ), both between models and in time, are difficult to

40 diagnose. We apply a neural network (NN) approach to address this issue within a group of models that participated in the Chemistry-Climate Model Initiative (CCMI). Analysis of the historical specified dynamics simulations performed for CCMI

indicates that the primary drivers of  $\tau_{CH_4}$  differences among ten models are the flux of UV light to the troposphere (indicated by the photolysis frequency JO1D)p mixing ratio of tropospheric ozone (O3), the abundance of nitrogen oxides (NOx=NO+NO2), 
[revised manuscript text omitted]

| ( | Moved (insertion) [2]                                                                                                                                                                                   |
|---|---------------------------------------------------------------------------------------------------------------------------------------------------------------------------------------------------------|
| ( | Deleted:                                                                                                                                                                                                |
| Ć | Deleted: s                                                                                                                                                                                              |
| ( | Deleted: with both                                                                                                                                                                                      |
| ( | Deleted: and                                                                                                                                                                                            |
| ( | Field Code Changed                                                                                                                                                                                      |
|   | Deleted: ; Duncan et al., 2003                                                                                                                                                                          |
|   | Deleted: . The former event likely impacted JO 1 D through the decrease in stratospheric O 3 that resulted (Aquila et al., 2013; Tie and Brasseur, 1995), while the latter |
|   | Deleted: Both effects will tend to increase the flux of UV light to the troposphere, increasing the primary production of OH and                                                                 |
|   | Moved up [2]: and decreasing $\tau_{CH_4}$ , as seen in Fig. 7.                                                                                                                                  |
| 2 | Deleted:                                                                                                                                                                                                |
|   | Deleted: the                                                                                                                                                                                            |
|   | Deleted: mentioned above was driven specifically by El Niño conditions.                                                                                                                          |
|   | Deleted: a                                                                                                                                                                                              |
| ( | Deleted: El Niño                                                                                                                                                                                        |
|   |                                                                                                                                                                                                         |

[revised manuscript text omitted]

**Moved (insertion) [1] Deleted: t**

Deleted: The response of OH to CH4 however, is in poor agreement between the two studies. The previously determined observation-based estimate of [OH]TROP trend due to CH4 was - $1.01\pm0.05$  % decade-1 while the CCMI model-based trend in  $\tau_{CH}$ , is only  $\pm 0.06 \pm 0.07$  % decade-1. On one hand, the treatment of CH4 as a normalized value within the NN analysis, as noted above in the discussion of Fig. 7, precludes a realistic estimate of the OH response to changes in CH4. Rather, the trend estimate calculated by the NN analysis of CCMI models represents the impact on OH of changes in the distribution of CH4 within the troposphere. Since the source regions of CH4 are not expected to change substantially over the 1980-2015 period, it is not surprising that the CCMI modelbased trend is small. Meanwhile, the other estimate of the [OH]TROP trend due to CH4 from Nicely et al. (2018) is not without limitations. As was acknowledged in that paper, the box model method used to estimate the sensitivities of OH to CH4 (among other species) is inherently inadequate for capturing complex coupling of chemical systems and downstream effects. For example, the box modelled sensitivity of OH to variations in CH4 were found for a range of latitude, pressure, and NOx values (since the latter determines whether CH4 oxidation consumes or regenerates OH radicals). To maintain the characteristics of those chemical regimes, then, O3 was input and held fixed in the box model simulation. As a result, especially in the relatively low-NOx conditions prevalent throughout much of the troposphere, an increase in CH4 would tend to consume OH without the corresponding increase in O3 expected to result from greater CH4 oxidation. That increase in tropospheric O3 would offset some of the OH loss by increasing primary production, a process that should be captured in a fully coupled chemistry-climate model like those participating in CCMI.

705 the further examination of the response of OH to CH40on the global scale, which is likely a large influencer of tropospheric OH abundance, as indicated in Nicely et al. (2018) and Holmes et al. (2013),

**5 Conclusions**

model's results.

We perform a neural network analysis of the monthly mean output from historical simulations of ten models that participated in CCMI for the purposes of understanding OH and  $\tau_{CH_4}$  differences and temporal trends. NNs are trained to reproduce OH

- 710 mixing ratios for a given model using 3-D fields of 12 OH precursor and sink parameters. Performing swaps of the NN inputs between models produces a quantitative estimate of the difference in *τCH4* that can be attributed to variations in the substituted variable. Among the ten models that we examine, on average, variations in JO1D, local O3, NOx, and chemical mechanisms account for the largest differences in *τCH4*. Model diversity in representations of H2O, CO, the partitioning of NOx, and HCHO is responsible for moderate OH differences, while isoprene, CH4, JNO2, overhead O3 column, and temperature account for little-to-no variation in OH. However, the relative importance of a particular variable is highly model-dependent, so any effort to improve the representation of OH within a given model should be guided by that particular
  - We also analyse time series of  $\tau_{CH_4}$  using the year 2000 NNs generated for the first half of the study. All models exhibit a downward trend in  $\tau_{CH_4}$  between 1980 and 2015, varying from -0.54 % decade-1 to -2.97 % decade-1 (average of -1.83 %
- 720 decade-1). Swaps of NN inputs are conducted between years rather than between models, so attributions of the factors influencing trends in  $\tau_{CH_4}$  are found for each model and then combined into a multi-model mean result. This analysis indicates that the largest contributors to the decreasing trend in  $\tau_{CH_4}$  are O3, JO1D, NOx, and H2O, while CO also imparts a large degree of interannual variability. Features due to strong ENSO events and associated biomass burning as well as the eruption of Mount Pinatubo are discernible in the time series of attributed variations in  $\tau_{CH_4}$ . In particular, the species CO,
- 725 H2O, and O3 instigate prominent responses during strong El Niño years. Finally, the attributed trends in  $\tau_{CH_4}$  from the NN analysis of CCMI model output are compared to trends in tropospheric mean OH concentration found previously in the observation-based study of Nicely et al. (2018). While the strong response of  $\tau_{CH_4}$  to increasing H2O over time appears to be a robust result, disagreement on the CH4 feedback on OH between the two studies highlights limitations in the approaches of both, in addition to more systemic issues in the community's ability to model CH4.
- 730 The NN and machine learning methods in general provide a valuable tool for performing insightful model intercomparisons of complex systems in a computationally-efficient manner. These approaches, however, must be undertaken with care to avoid erroneous results and recognition of their limitations. At present, we have devised a method to identify the drivers of OH variations, whether between models or between years, at coarse temporal resolution. Much future work is needed, though: observations must be incorporated to introduce a ground truth element to this analysis in a manner that either adjusts

**20**

**Deleted: ,**

**Deleted: known as the CH4 feedback,**

**Moved up [1]:** the results depicted in Fig. 10 show relatively robust findings regarding the responses of  $[OH]^{TROP}$  and  $\tau_{CH_4}$  to the factors examined through two independent studies.

**Deleted:** This topic has been of interest for some time (Holmes et al., 2013; Prather et al., 2001), though the necessity of providing boundary conditions for surface CH4 rather than fluxes in models hampers our ability to realistically simulate CH4. Regardless, a model approach using fully-coupled tropospheric chemistry, such as that performed by Holmes et al. (2013) for three CTMs, would provide a more direct measure of the CH4 feedback on OH than both approaches depicted in Fig. 10. Except for trends attributed to CH4,

for or avoids disconnects between coarse versus local/instantaneous spatiotemporal scales and appropriately accounts for measurement uncertainty; an analysis of model output with much higher temporal frequency is needed to identify exactly

750 where model differences in chemical mechanisms lie; and subsequent studies of why the various OH precursor and sink fields differ are required to make this analysis of greatest utility for improving model representations of  $\tau_{CH_4}$ . While these challenges are significant, they are not insurmountable, especially as machine learning and other advanced statistical analysis techniques continue to be developed and honed.

**Data Availability**

- 755 All output from most of the models that participated in CCMI is available at the Centre for Environmental Data Analysis (CEDA), the Natural Environment Research Council's Data Repository for Atmospheric Science and Earth Observation, at <a href="http://data.ceda.ac.uk/bade/wcrp-ccmi/data/CCMI-1/output">http://data.ceda.ac.uk/bade/wcrp-ccmi/data/CCMI-1/output</a>. WACCM and CAM4-Chem output for CCMI is available for download at <a href="http://www.earthsystemgrid.org">http://www.earthsystemgrid.org</a>. For instructions for access to both archives see <a href="http://blogs.reading.ac.uk/ccmi/bade-data-access">http://blogs.reading.ac.uk/ccmi/bade-data-access</a>. 
[revised manuscript text omitted]